# Urolithin A Protects Porcine Oocytes from Artificially Induced Oxidative Stress Damage to Enhance Oocyte Maturation and Subsequent Embryo Development

**DOI:** 10.3390/ijms26073037

**Published:** 2025-03-26

**Authors:** Wen Shi, Chaobin Qin, Yanyan Yang, Xiaofen Yang, Yizhen Fang, Bing Zhang, Dong Wang, Wanyou Feng, Deshun Shi

**Affiliations:** 1Guangxi Key Laboratory of Animal Breeding & Disease Control and Prevention, College of Animal Science and Technology, Guangxi University, Nanning 530004, China; 2118401004@st.gxu.edu.cn (W.S.); 2018401005@st.gxu.edu.cn (C.Q.); 2218401005@st.gxu.edu.cn (Y.Y.); 2218401004@st.gxu.edu.cn (X.Y.); 2218391010@st.gxu.edu.cn (Y.F.); 2118301041@st.gxu.edu.cn (B.Z.); 2218301027@st.gxu.edu.cn (D.W.); 2School of Environmental and Life Sciences, Nanning Normal University, Nanning 530001, China

**Keywords:** urolithin A, oxidative stress, mitochondrial function, autophagy, oocyte quality, meiotic maturation, embryo development

## Abstract

Both the livestock and biomedical fields require a large supply of high-quality mature oocytes. However, the in vitro maturation (IVM) process often leads to an accumulation of reactive oxygen species (ROS), which can cause defects in oocyte meiosis and embryo development, ultimately compromising oocyte quality. Urolithin A (UA), known for its antioxidant properties, has not been thoroughly investigated for its potential to mitigate the negative effects of oxidative stress during the in vitro culturing of oocytes, and its underlying mechanism is not well understood. In this study, an in vitro oxidative stress model was established using porcine oocytes treated with H_2_O_2_, followed by exposure to varying concentrations of UA. The results revealed that 30 μM UA significantly improved both the quality of oocyte culture and the developmental potential of the resulting embryos. UA was found to enhance oocyte autophagy, reduce oxidative stress-induced mitochondrial damage, and restore mitochondrial function. Additionally, it lowered ROS and DNA damage levels in the oocytes, maintained proper spindle/chromosome alignment and actin cytoskeleton structure, promoted nuclear maturation, prevented abnormal cortical granule distribution, and supported oocyte cytoplasmic maturation. As a result, UA alleviated oxidative stress-induced defects in oocyte maturation and cumulus cell expansion, thereby improving the developmental potential and quality of parthenogenetic embryos. After supplementation with UA, pig parthenogenetic embryo pluripotency-related genes (*Nanog* and *Sox2*) and antiapoptotic genes (*Bcl2*) were upregulated, while proapoptotic genes (*Bax*) were downregulated. In conclusion, this study suggests that adding UA during IVM can effectively mitigate the adverse effects of oxidative stress on porcine oocytes, presenting a promising strategy for enhancing their developmental potential in vitro.

## 1. Introduction

Porcine are one of the most important livestock species worldwide, serving as the largest source of meat and providing humans with a stable and high-quality protein supply. Moreover, due to their high physiological and genetic similarity to humans, porcine are widely used in constructing human disease models, xenotransplantation research, transgenic animal studies, and drug development and toxicity testing [1,2].

Therefore, it is imperative to increase the efficiency of the in vitro production (IVP) of porcine embryos for not only agricultural production but also for biomedical research purposes [3]. The impairment of oocyte quality during in vitro maturation (IVM) is a critical factor associated with the failure of assisted reproductive technologies (ARTs) [4,5,6], since its quality is the main determinant for the embryo’s developmental potential after fertilization [7,8,9]. While significant progress has been made in culturing oocytes for IVM, compared with the mature oocytes in vivo, in vitro mature porcine oocytes still have some problems, such as poor quality [10,11] and the higher polyspermic fertilization rate of in vitro fertilization (IVF) [12,13,14], so the process is far from meeting the standards required for practical applications. A major contributor to this diminished quality is the generation of reactive oxygen species (ROS) during the IVM process [15]. The IVM process leads to an increase in ROS; furthermore, oocytes cultured in vitro lack the protection of antioxidant system in vivo, consequently resulting in them being more susceptible to free radical damage [15,16]. This disruption of the redox balance can lead to structural and functional damage, including mitochondrial dysfunction, cytoskeletal abnormalities, chromosomal anomalies, irregular cortical granule distribution, and DNA damage [10,17]. These issues hinder successful embryo development. Recent studies show that antioxidant treatments can restore the intracellular redox balance, mitigating oxidative stress during the IVM process [18,19,20]. This not only prevents oxidative damage but also improves the developmental potential and quality of the blastocysts. Supplementing antioxidants in vitro has proven to be an effective strategy for improving oocyte maturation quality and efficiency, addressing one of ART’s major challenges [16,21].

Urolithin A (UA), a member of the urolithin family, is a metabolite derived from ellagic tannins (ETs) and ellagic acid (EA) through gut microbial activity [22]. Research has demonstrated that UA possesses a range of biological functions, including antioxidant properties [23,24], the maintenance of mitochondrial function [23], anti-aging effects [25], the regulation of estrogen levels [26], and anti-inflammatory activity [27]. A previous study incorporated UA into bovine IVF sperm capacitation and fertilization medium, and their findings revealed that UA reduced oxidative stress [28] in bovine sperm while enhancing mitochondrial function, which in turn has a positive impact on improved sperm vitality and quality and on the number of embryos produced in vitro [29]. Yi Hong et al. reported that UA alleviated ovarian oxidative stress in mouse embryos (14.5 days post-coitus, dpc) induced by exogenous toxins (HT-2), partially reversing the delayed meiotic progression in oocytes [30]. Post-ovulatory aging and maternal aging-related oocyte aging significantly impair female fertility, but UA is reported to restore mitochondrial membrane potential in aged bovine oocytes and to promote subsequent embryonic development [31]. This discovery indicates that UA offers a promising therapeutic strategy for preventing or delaying gamete aging in ARTs, enhancing blastocyst formation and improving fertility outcomes. Moreover, UA has also been proven to regulates mitochondrial biogenesis and eliminates damaged mitochondria via mitophagy, thereby maintaining mitochondrial activity and delaying both cellular and organismal aging [25]. UA supplementation has been found to have a positive effect on a variety of tissues and age-related decline phenotypes, as another study showed that UA enhances mitochondrial function in aging *C. elegans* through mitophagy, thereby improving oocyte quality and extending reproductive function [28]. Although research indicates that UA has multiple biological activities, and some clinical studies highlight its beneficial effects on aging and age-related diseases, particularly in muscles [25], the brain [32], joints [33], kidneys [34], and the metabolic system [35], its effects on reproduction remain underexplored. The impact of UA on pig oocyte IVM and embryonic development potential is still not well understood.

In this study, we constructed an in vitro oxidative stress model for oocytes using H_2_O_2_ and investigated the positive effects of UA treatment on oocytes IVM, by evaluating cumulus cell expansion, cortical granule localization, ROS levels, spindle organization, chromosome alignment, DNA damage levels, mitochondrial content, ATP levels, and mitochondrial membrane potential. Additionally, we further evaluated the developmental competence and quality of blastocysts (BLs) obtained following parthenogenesis to determine if UA has a beneficial impact on oocytes subjected to oxidative damage. Our research may provide new strategies for enhancing the production efficiency of the IVM of pig oocytes and for promoting the development of assisted reproductive technologies.

## 2. Results

### 2.1. Establishment of an Artificially Induced Oxidative Stress Model in Porcine Oocytes

To establish an oxidative stress model in porcine oocytes, cumulus–oocyte complexes (COCs) exhibiting multiple layers of cumulus cells and homogeneous cytoplasm were exposed to varying concentrations (0, 50, 100, 200, and 400 μM) of H_2_O_2_ in maturation medium for 30 min, followed by cultivation in standard medium until they reached the MII stage. Upon the completion of IVM, multiple parameters were evaluated across all experimental groups, including cumulus expansion, the first polar body extrusion (PBE) rate, and subsequent developmental competence following parthenogenetic activation. Cumulus expansion was quantitatively assessed using the Cumulus Expansion Index (CEI) criteria, with representative images presented in Figure 1A. In comparison with the control group, H_2_O_2_ treatment significantly impaired cumulus cell expansion during oocyte maturation (control 2.15 ± 0.12, *n* = 265 vs. 50 μM 1.37 ± 0.04, *n* = 299, 100 μM 1.31 ± 0.03, *n* = 275, 200 μM 1.16 ± 0.05, *n* = 288, and 400 μM 0.90 ± 0.07, *n* = 259, *p* < 0.0001), demonstrating a concentration-dependent inhibitory effect (Figure 1B,C). An analysis of polar body extrusion following maturation revealed that H_2_O_2_ treatment induced a dose-dependent reduction in the first PBE rates compared to the control group (Figure 1D, Appendix A), with 200 μM and 400 μM H_2_O_2_ exhibiting the most pronounced inhibitory effects (control 83.68 ± 4.12%, *n* = 201 vs. 200 μM 53.88 ± 2.19%, *n* = 225 and 400 μM 33.19 ± 2.89%, *n* = 213, *p* < 0.0001; Appendix A). Following maturation, oocytes from each experimental group underwent parthenogenetic activation, and both cleavage and day 6 blastocyst rates were documented. Treatment with 200 μM and 400 μM H_2_O_2_ significantly compromised developmental competence compared to the control group, with 400 μM H_2_O_2_ resulting in a markedly reduced cleavage rate of 14.58% and minimal blastocyst formation (3.45%) (Figure 1E,F). Based on a comprehensive assessment of cumulus expansion, the first PBE rates, and developmental potential following parthenogenetic activation, 200 μM H_2_O_2_ was identified as the optimal concentration for establishing an in vitro oxidative stress model in porcine oocytes.

### 2.2. Effects of UA on Porcine Oocyte Maturation and Developmental Competence Under Artificially Induced Oxidative Stress Damage

Our results showed that the addition of UA (20 and 40 μM) during porcine conventional oocytes IVM could slightly promote the PBE rate (*p* > 0.05). Subsequent embryonic development results revealed that 20 μM UA could significantly increase the parthenogenetic embryo developmental competence of porcine oocytes (control 31.01 ± 0.85%, *n* = 178 vs. 20 μM 36.23 ± 1.81%, *n* = 150, *p*  <  0.05, Figure 2 and Appendix A). To investigate the effects of varying UA concentrations on the IVM of oxidatively stressed porcine oocytes and their subsequent developmental competence following parthenogenetic activation, oxidatively stressed oocytes were cultured in maturation medium supplemented with different concentrations of UA (0, 5, 10, 15, 30, 60, and 120 μM). After culture maturation, the expansion of cumulus cells during oocyte maturation was restored to different degrees after adding different concentrations of UA compared with the H_2_O_2_-treated group (0 μM group), and the effect of the 30 μM UA-supplemented group was the best (0 μM 1.06 ± 0.093%, *n* = 233 vs. 30 μM 1.74 ± 0.103%, *n* = 259, *p* < 0.0001, Figure 3A,B). Following culture maturation, an analysis of the first PBE rates demonstrated significant improvements in groups treated with 5, 10, 15, 30, and 60 μM UA compared to the 0 μM group (0 μM 51.67, *n* = 274 ± 3.33% vs. 5 μM 68.15 ± 2.94%, *n* = 179, *p* < 0.001, 0 μM 51.67 ± 3.33%, *n* = 274 vs. 10 μM 76.15 ± 1.26%, *n* = 173, 15 μM 80.94 ± 1.15%, *n* = 184 and 30 μM 84.85 ± 1.18%, *n* = 281, *p* < 0.0001, 0 μM 51.67 ± 3.33% vs. 60 μM 64.50 ± 3.03%, *n* = 243, *p* < 0.05; Figure 3C). Notably, the 15 and 30 μM groups showed comparable PBE rates to the control group (*p* > 0.05), and the 30 μM group had the best effect on promoting polar body extrusion (Appendix A). Upon the completion of maturation, oocytes from each experimental group underwent parthenogenetic activation for developmental assessment. Cleavage rates were significantly enhanced in groups treated with 5–60 μM UA compared to the 0 μM group (0 μM 56.80 ± 1.04%, *n* = 274 vs. 5 μM 70.38 ± 0.49%, *n* = 179, 10 μM 74.07 ± 1.20%, *n* = 173, *p* < 0.001, 0 μM 56.80 ± 1.04%, *n* = 274 vs. 60 μM 70.01 ± 2.61%, *n* = 243, *p* < 0.01, 0 μM 56.80 ± 1.04%, *n* = 274 vs. 15 μM 78.74 ± 0.97%, *n* = 184 and 30 μM 81.92 ± 1.40%, *n* = 227, *p* < 0.0001), with the 30 μM treatment showing optimal results approximating control levels (Figure 3D, Appendix A). There was a downward trend in the cleavage capacity of the 120 μM (*n* = 163) UA treatment (*p* > 0.05). As shown in Figure 3E, day 6 blastocyst rates increased to various degrees after adding different concentrations of UA, with the 30 μM group achieving near-control levels. Based on a comprehensive evaluation of cumulus expansion, the first PBE rates, and developmental competence following parthenogenetic activation, supplementation with 30 μM UA was determined to optimally restore the maturation and developmental capacity of oxidatively damaged oocytes. Consequently, 30 μM UA was established as the optimal concentration for subsequent experimental investigations. In addition, we attempted to examine the effect of UA (30 μM) on the embryonic development of IVF embryos. The result showed that the addition of UA slightly increased the development of porcine IVF embryos under oxidative stress (H_2_O_2_ 2.90 ± 1.01%, *n* = 288 vs. H_2_O_2_+UA 3.93 ± 0.56%, *n* = 288, *p* >0.05, Appendix A).

### 2.3. UA Supplementation Enhances Antioxidant Capacity in Porcine Oocytes with Artificially Induced Oxidative Stress Damage

To investigate the underlying molecular mechanisms of the UA-mediated restoration of oocyte maturation competence and subsequent parthenogenetic developmental potential following oxidative damage, intracellular ROS levels were quantitatively assessed by the oxidation-sensitive fluorescent probe dichlorofluorescein diacetate (DCFHDA) in MII stage oocytes post-maturation. The fluorescence microscopy analysis and quantitative assessment of signal intensity results are shown in Figure 4A,B. While H_2_O_2_ exposure induced a significant elevation in intracellular ROS accumulation, UA treatment effectively mitigated this oxidative burden, though not completely to the control levels. Expression analysis of key antioxidant defense genes (*SOD1*, *CAT*, and *GPX4*) by RT-qPCR demonstrated that UA supplementation successfully restored the transcriptional levels of these antioxidant enzymes following oxidative stress. Notably, *CAT* expression was significantly upregulated, exceeding the control group levels (*p* < 0.01).

### 2.4. UA Supplementation Attenuates DNA Damage in Artificially Induced Oxidative Stress-Damaged Porcine Oocytes

In light of the well-documented relationship between elevated ROS levels and DNA damage, we employed γ-H2A.X (the key regulator of DNA damage) immunofluorescence staining to assess the extent of genomic damage. Quantitative analysis of fluorescence imaging, as depicted in Figure 5A,B, revealed that H_2_O_2_ exposure significantly enhanced γ-H2A.X immunofluorescence intensity relative to control oocytes. Although UA supplementation significantly attenuated this effect, residual γ-H2A.X signals remained detectably higher than baseline controls (control 40.19 ± 1.32 vs. H_2_O_2_ 65.82 ± 1.35, H_2_O_2_+UA 54.80 ± 1.31, *p* <0.0001). These observations demonstrate that while oxidative stress induced substantial DNA damage in oocytes, UA treatment effectively counteracted this deleterious effect.

### 2.5. UA Supplementation Restores Spindle/Chromosome Defects in Artificially Induced Oxidative Stress-Damaged Porcine Oocytes

The integrity of spindle assembly and chromosomal alignment represents a critical determinant of oocyte maturation quality and subsequent developmental competence. Therefore, we examined the potential protective effects of UA against oxidative stress-induced disruption of these essential cellular structures. The results are shown in Figure 6 through high-resolution immunofluorescence analysis. By employing α-tubulin antibody for spindle visualization and Hoechst counterstaining for chromosomal detection, we observed that control oocytes consistently displayed well-defined barrel-shaped spindles with properly organized microtubular networks and precisely aligned chromosomes at the metaphase plate. H_2_O_2_ treatment significantly elevated the frequency of structural abnormalities in both spindle organization and chromosomal alignment. Significantly, UA supplementation effectively preserved normal cytoskeletal architecture and chromosomal configuration under oxidative stress conditions (control 17.93  ±  2.89% vs. H_2_O_2_ 45.64  ±  1.01%, *p*  <  0.001; control 17.93  ±  2.89% vs. H_2_O_2_+UA 31.43 ±  2.50, *p*  <  0.05, spindle; control 20.27  ±  2.16% vs. H_2_O_2_ 46.66  ±  1.22%, *p*  <  0.001; control 20.27  ±  2.16% vs. H_2_O_2_+UA 34.93 ± 2.15 *p*  <  0.01, chromosome).

### 2.6. UA Supplementation Rescues Actin Polymerization Abnormalities in Artificially Induced Oxidative Stress-Damaged Porcine Oocytes

The actin cytoskeleton plays a key role in nuclear positioning, spindle migration, and anchoring, and in polar body extrusion to promote meiotic progression in mammalian oocytes. Therefore, we used phalloidin to visualize the changes in actin polymerization in H_2_O_2_-treated oocytes exposed to UA. Immunofluorescence imaging and quantitative data revealed that actin integrity in porcine oocytes was impaired by oxidative stress. The results are shown in Figure 7A,B. In the control group, actin signals were strongly distributed on the plasma membrane, as shown by the plot of fluorescence intensity along a line drawn across the oocytes, while actin filaments on the membrane displayed a significantly weakened signals in the H_2_O_2_ group (control 65.43  ±  0.56, *n* = 78 vs. H_2_O_2_ 30.65  ±  0.55, *n* = 75, *p*  <  0.0001), but recovered in UA-supplemented oocytes (H_2_O_2_ 30.65  ±  0.55, *n* = 75 vs. H_2_O_2_+UA 66.75  ±  0.52, *n* = 81, *p*  <  0.0001). Consistently, the quantification of actin signals on the entire plasma membrane also verified that UA supplementation mitigated the defects in actin dynamics in postovulatory aged oocytes (Figure 7C). We conclude from these observations that UA maintains the actin cytoskeleton during oxidative stress to protect the oocytes’ integrity.

### 2.7. UA Supplementation Improves Mitochondrial Function in Artificially Induced Oxidative Stress-Damaged Porcine Oocytes

Under physiological conditions, intracellular ROS levels maintain homeostatic balance; however, excessive ROS accumulation can significantly impair mitochondrial function. In oocytes, abundant mitochondria serve as essential energy providers for various cellular processes, with mitochondrial activity directly correlating with oocyte maturation competence and subsequent embryonic developmental potential. Mitochondrial content was evaluated using the MitoTracker fluorescent probe across three experimental groups: control, H_2_O_2_-treated, and H_2_O_2_+UA-treated mature oocytes. Quantitative analysis revealed that H_2_O_2_-treated oocytes exhibited significantly diminished MitoTracker fluorescence intensity compared to controls (Figure 8A,B). Notably, UA supplementation significantly enhanced MitoTracker fluorescence intensity in oxidatively stressed oocytes (Figure 8A,B), indicating restored mitochondrial content during oocyte maturation (control 34.36 ± 0.46 vs. H_2_O_2_ 26.25 ± 0.80, *p * <  0.0001; control 34.36 ± 0.46 vs. H_2_O_2_+UA 30.51 ± 0.44, *p * <  0.001). Mitochondrial membrane potential (mMP), a prerequisite for oxidative phosphorylation and ATP generation, is crucial for maintaining cellular bioenergetics and physiological functions. Using JC-1 fluorescent probe analysis, we assessed mMP across treatment groups, which yields green or red fluorescence, to distinguish cells with high or low mitochondria potential, respectively. The mitochondria with a high membrane potential fluoresced red, whereas those with a low membrane potential fluoresced green. H_2_O_2_ exposure significantly reduced the red/green fluorescence ratio compared to controls, while UA supplementation markedly ameliorated this reduction (Figure 8D,E). This suggests that UA treatment effectively restored mitochondrial membrane potential towards physiological levels in oxidatively stressed oocytes. As primary cellular ATP generators, mitochondrial functionality was further assessed through ATP content measurement. Quantitative analysis demonstrated significantly depleted ATP levels in H_2_O_2_-treated oocytes compared to controls (Figure 8C). UA supplementation effectively restored intracellular ATP content (control 2.46 ± 0.08 vs. H_2_O_2_ 1.70 ± 0.03, *p * <  0.001; control 2.46 ± 0.08 vs. H_2_O_2_+UA 2.04 ± 0.05, *p * <  0.01). Expression analysis by RT-qPCR of mitochondrial function-related genes revealed that oxidative stress significantly downregulated mitochondrial function-related genes compared to controls, with *PGC-1α* showing particularly marked reduction (Figure 8F). UA supplementation significantly upregulated these mitochondrial-related gene expressions. Mitochondria damaged by oxidative stress have insufficient ability to provide the amount of ATP needed to sustain metabolic reactions, and the cell activates autophagy to rapidly degrade the damaged components and put them back into a new cycle for use. Compared with the control group, the expression levels of autophagy-related genes (*Beclin*, *LC3B* and *p62*) showed a downward trend after oocytes were damaged by oxidative stress, and the expression levels of *LC3B* (*p*  <  0.01) and *p62* (*p * <  0.0001) significantly decreased compared with the control group. The level of autophagy-related genes in oocytes was significantly increased by adding UA (Figure 8G). Collectively, these findings provide compelling evidence that UA supplementation effectively enhances oocyte autophagy level and ameliorates oxidative stress-induced mitochondrial dysfunction in porcine oocytes.

### 2.8. UA Rescues Cortical Granule Distribution in Artificially Induced Oxidative Stress-Damaged Porcine Oocytes

Cortical granules constitute highly specialized secretory vesicles exclusively found in mammalian oocytes and function as essential mediators of the polyspermy block during fertilization. Their precise spatial distribution pattern is widely acknowledged as a fundamental marker of cytoplasmic maturation competence. Through LCA-FITC (lens culinaris agglutinin–fluorescein labeled) labeling, we conducted comprehensive analyses of cortical granule (CG) distribution patterns and density in mature oocytes across the experimental groups. Quantitative fluorescence imaging analysis revealed that oxidative stress treatment significantly diminished CG signals within the subcortical region of oocytes (Figure 9A,B). Notably, UA supplementation effectively mitigated this reduction in CG signal intensity, indicating UA supplementation proper restoration of cortical granule distribution (control 37.22 ± 0.54 vs. H_2_O_2_ 13.81 ± 0.80, H_2_O_2_+UA 23.85 ± 0.48, *p * <  0.0001).

### 2.9. UA Supplementation Enhances the Quality of Parthenogenetic Embryo Development in Artificially Induced Oxidative Stress-Damaged Porcine Oocytes

The aforementioned findings demonstrated that oxidative stress significantly compromises oocyte maturation competence and cellular integrity. To further validate the protective effects of UA against oxidative stress in oocytes, we evaluated early parthenogenetic embryo quality parameters. Previous studies have established total cell number as a reliable indicator of blastocyst developmental competence. As demonstrated in Section 2.2, H_2_O_2_ exposure significantly impaired parthenogenetic embryo development rates. Blastocysts were collected from all experimental groups following 6 days of in vitro culturing, subsequently fixed and stained with Hoechst, and then subjected to fluorescence microscopic analysis for a quantitative assessment of blastocyst total cell numbers. As illustrated in the figure (Figure 10A,B), control group blastocysts exhibited superior morphological characteristics with expanded size and significantly higher cell numbers compared to both H_2_O_2_-treated and H_2_O_2_+UA groups. H_2_O_2_ exposure substantially compromised parthenogenetic embryo developmental potential, manifested as significantly reduced blastocyst total cell numbers. Notably, UA supplementation significantly enhanced blastocyst total cell numbers compared to the H_2_O_2_-treated group, demonstrating its effective amelioration of oxidative stress-induced developmental impairment in porcine parthenogenetic embryos (control 49.40 ± 1.37 vs. H_2_O_2_ 31.77 ± 2.10, *p * <  0.0001; control 49.40 ± 1.37 vs. H_2_O_2_+UA 38.09 ± 1.46, *p * <  0.001). In addition, the mRNA levels of genes related to the developmental potential of parthenogenetic blastocysts *(Nanog* and *Sox2*) and apoptosis (*Bax* and *Bcl2*) were measured. The mRNA expression of the pluripotency genes *Nanog* (control 1.07 ± 0.02 vs. H_2_O_2_ 0.70 ± 0.00, *p * <  0.0001) and *Sox2* (control 0.9 ± 0.03 vs. H_2_O_2_ 0.65 ± 0.02, *p * <  0.001) of the blastocysts in the H_2_O_2_ group was significantly lower than that of the control group. The addition of urolithin A improved the expression of *Nanog* (H_2_O_2_ 0.70 ± 0.00 vs. H_2_O_2_+UA 0.84 ± 0.01, *p * <  0.001) and *Sox2* (H_2_O_2_ 0.65 ± 0.02 vs. H_2_O_2_+UA 1.02 ± 0.03, *p * <  0.0001) following oxidative stress. Oxidative stress significantly decreased the expression of antiapoptotic gene (*Bcl2*, control 1.00 ± 0.04 vs. H_2_O_2_ 0.68 ± 0.04, *p * <  0.0001) and increased the expression of proapoptotic genes (*Bax*, control 1.00 ± 0.04 vs. H_2_O_2_ 2.41 ± 0.11, *p * <  0.0001), and the addition of UA significantly alleviated the level of apoptosis in parthenogenetic embryos (*Bcl2*, H_2_O_2_ 0.68 ± 0.04 vs. H_2_O_2_+UA 1.09 ± 0.05, *p * <  0.001; *Bax*, H_2_O_2_ 2.41 ± 0.11 vs. H_2_O_2_+UA 1,41 ± 0.01, *p * <  0.0001).

## 3. Discussion

The production of high-quality oocytes represents a critical component in successful in vitro embryo development. However, oocyte maturation failure during IVM can severely impede embryo development, implantation, and pregnancy maintenance [36,37]. Despite numerous studies aimed at improving IVM outcomes [38,39,40], IVM rates remain inferior to those observed in vivo [41]. Among the multiple factors affecting IVM oocyte quality, elevated oxidative stress has been identified as a primary contributor to maturation failure [42]. Notably, higher levels of oxidative stress are detected during IVM compared to in vivo conditions, leading to compromised oocyte maturation [43]. ROS are a class of metabolic byproducts formed during cell metabolism, which mainly include superoxide anions (O^2−^), hydroxyl radicals (HO^−^), alkoxy (RO^−^, peroxy (RO^2−^), and other non-radicals, such as H_2_O_2_ [44]. Excessive ROS concentrations exhibit cytotoxicity and induce oocyte damage, including DNA damage and apoptosis, ultimately severely compromising oocyte quality. H_2_O_2_ functions as a key second messenger and versatile physiological signaling molecule involved in metabolic regulation and stress responses [42]. H_2_O_2_ is recognized as the major ROS in redox regulation of biological activities, and it serves as the most direct and effective mediator of cellular oxidative stress [45,46]. Consequently, we selected H_2_O_2_ treatment to establish an in vitro oxidative stress model in porcine oocytes to investigate UA’s protective effects against oxidative stress-induced meiotic damage. Various antioxidant therapies have been proposed to improve oocyte quality [47,48,49]. While previous studies have demonstrated UA’s diverse biological activities and potential role in improving oocyte quality, its specific mechanisms and effects remain unexplored [30,31].

In this investigation, we initially evaluated the impact of varying H_2_O_2_ concentrations on porcine oocyte maturation through a comprehensive assessment of multiple parameters: cumulus cell expansion, first polar body extrusion, day 2 cleavage rates, and day 6 blastocyst rates following parthenogenetic activation. Our results demonstrated that H_2_O_2_ exposure significantly suppressed cumulus cell expansion in a concentration-dependent manner. Specifically, treatments with 200 μM and 400 μM H_2_O_2_ significantly diminished polar body extrusion rates, cleavage rates, and blastocyst formation rates. Notably, exposure to 400 μM H_2_O_2_ severely compromised oocyte developmental competence, resulting in minimal blastocyst development from parthenogenetically activated embryos. Based on these observations, 200 μM H_2_O_2_ was established as the optimal concentration for developing an in vitro oxidative stress model in porcine oocytes, aligning with previous findings reported by Chen Pan et al. [50]. Before exploring the effect of UA on porcine oocyte maturation and subsequent parthenogenetic embryo development under artificially induced oxidative stress model, we first tested the potential effect of UA on porcine oocytes conventional in vitro maturation. The results showed that the addition of UA (20 and 40 μM) during porcine conventional oocytes IVM could slightly promote the PBE rate (*p* > 0.05), while 20 μM UA could significantly increase parthenogenetic embryos’ developmental competence of porcine oocytes; therefore, UA has a potential positive effect on improving the maturation and embryonic ability of porcine oocytes. We subsequently investigated the protective effects of various UA concentrations on oxidatively stressed porcine oocytes during IVM and their subsequent parthenogenetic developmental potential. Our results revealed that supplementation with 5–30 μM UA demonstrated dose-dependent mitigation of oxidative stress-induced impairments in both oocyte maturation and embryo development, with 30 μM UA exhibiting optimal protective efficacy. Supplementation with 30 μM UA effectively restored multiple developmental parameters, including cumulus cell expansion, first polar body extrusion, and parthenogenetic embryo cleavage to levels comparable with the control group. Further analysis of blastocyst development confirmed that UA supplementation significantly enhanced the developmental potential of oxidatively stressed oocytes, although a modest gap persisted compared to control levels.

Our findings demonstrated that oxidatively stressed porcine oocytes exhibited significantly compromised developmental competence. To elucidate the mechanisms underlying this quality decline, we investigated various cellular parameters. Oocytes cultured in vitro demonstrate particular vulnerability to free radical-induced structural and functional damage [17]. UA, with its inherent antioxidant properties, significantly enhanced antioxidant gene expression and attenuated intracellular ROS levels in oxidatively stressed oocytes. Excessive accumulation of free radicals beyond the antioxidant system’s clearance capacity results in ROS-mediated damage to cellular biomolecules, including DNA, proteins, and lipids [51,52]. Our analyses revealed that H_2_O_2_ exposure significantly increased DNA damage in oocytes, while supplementation with 30 μM UA effectively ameliorated this oxidative stress-induced genomic damage. During oocyte meiotic and nuclear maturation, spindle assembly, chromosomal alignment, and the dynamics of the actin cytoskeleton represent critical developmental events. H_2_O_2_ treatment severely disrupted these processes during oocyte maturation. Notably, UA supplementation restored these cytoskeletal structural defects, demonstrating its capacity to normalize oxidative damage-induced cytoskeletal abnormalities. Sustained oxidative stress induces substantial mitochondrial damage, significantly compromising oocyte quality and reproductive outcomes [53,54]. In our study, while H_2_O_2_ treatment significantly impaired mitochondrial function, UA supplementation effectively mitigated this damage, as evidenced by improvements in multiple parameters, including mitochondrial content, ATP levels, membrane potential, and mitochondria-related gene expression. Autophagy, a cellular quality control mechanism targeting abnormal proteins, lipids, and organelles, facilitates cellular adaptation to adverse conditions and maintains homeostasis [55]. Severe oxidative stress triggers mitochondrial damage [56], which can be counteracted through three autophagic pathways, namely Beclin/VPS34/Atg14, AMPK/mTOR, and p62/Keap1/Nrf2 [56]. In this study, UA treatment significantly upregulated autophagy-related gene expression (*Beclin*, *LC3B*, and *p62*) in oxidatively stressed oocytes, suggesting its role in reducing the level of oxidative stress in oocytes by enhancing oocyte autophagy, thereby restoring the level of mitochondria and providing assistance for oocyte maturation and subsequent embryonic development. Cortical granules, specialized secretory vesicles in unfertilized oocytes, serve as critical indicators of cytoplasmic maturation [57]. Our analyses revealed significantly reduced CG numbers in the subcortical region of H_2_O_2_-treated oocytes, indicating disrupted CG distribution, which was effectively restored by UA supplementation.

Quantitative results of blastocysts mRNA expression showed that after supplementation with UA, pig parthenogenetic embryo pluripotency-related genes (*Nanog* and *Sox2*) and antiapoptotic genes (*Bcl2*) were upregulated, while proapoptotic genes (*Bax*) were downregulated. The total cell number of blastocysts is one of the important indexes to evaluate the quality of blastocysts [58,59]. Quantitative assessment of blastocyst total cell numbers demonstrated that UA treatment significantly enhanced cellular proliferation, indicating improved developmental competence. Collectively, these findings support UA supplementation’s therapeutic potential in ameliorating H_2_O_2_-induced defects in porcine oocyte maturation and developmental capacity, although further mechanistic studies are warranted.

## 4. Materials and Methods

### 4.1. Oocyte Collection and Maturation In Vitro

Porcine ovaries were collected from a local slaughterhouse and immediately placed in 30 °C physiological saline supplemented with penicillin G and streptomycin sulfate, followed by transportation to the laboratory within 2 h. Using disposable syringes, cumulus–oocyte complexes (COCs) were carefully aspirated from follicles measuring 3–6 mm in diameter. IVM was performed on oocytes with a compact cumulus cell in TCM-199 (ThermoFisher Scientific, Waltham, MA, USA) supplemented with 5 g/mL insulin, 10 ng/mL EGF, 0.2 mM pyruvate, 10% porcine follicular fluid, 0.6 mM cysteine, 25 μg/mL kanamycin, and 10 IU/mL eCG and hCG. The medium was overlaid with mineral oil, and 20–30 germinal vesicle (GV) stage COCs were cultivated at 38.5 °C for 44 h in an incubator with a humidified atmosphere containing 5% CO_2_.

### 4.2. Artificially Induced Oxidative Stress-Damage and Urolithin A Supplementation

H_2_O_2_ (hydrogen peroxide 30%, Nanjing Reagent, ACS7722–84–1, Nanjing, China) was deliquated in mature medium into concentrations of 50, 100, 200 or 400 μM, respectively. The GV oocytes were cultured in the medium with H_2_O_2_ working concentration for 30 min, then removed and washed with media three times and prepared along with the other groups. Each group of oocytes was treated with H_2_O_2_ to provide exogenous oxidative stress and then transferred to medium supplemented with UA.

UA (MedChemExpress, Shanghai, China) was initially dissolved in dimethyl sulfoxide (DMSO) to prepare a concentrated stock solution of 100 mM and stored at −20 °C. Prior to experimentation, the stock solution was serially diluted with maturation medium to achieve final concentrations of 5, 10, 15, 30, 60, and 120 μM. Following oxidative stress induction with hydrogen peroxide, each experimental group of oocytes was subsequently transferred to a culture medium supplemented with the respective concentrations of UA in accordance with the experimental design.

### 4.3. Statistical Analysis of the Cumulus Cell Expansion Index

After culturing COCs for 42–44 h, the CEI was calculated based on the COCs morphology, as previously described [60]. The cumulus expansion of COCs was divided into five degrees: degree 0 indicates no cumulus expansion; degree 1 reflects an expansion confined within the outermost layer of 1–2 cumulus cells; degree 2 indicates a radial expansion of the outer cumulus cells, resulting in relatively fluffy COCs; degree 3 indicates that only the rest of the oocyte expanded, while the corona radiata did not; degree 4 indicates that all cumulus cells have expanded. The CEI value was calculated using the following formula: CEI = [(number of degree 0 oocytes × 0) + (number of degree 1 oocytes × 1) + (number of degree 2 oocytes × 2) + (number of degree 3 oocytes × 3) + (number of degree 4 oocytes × 4)]/total number of oocytes.

### 4.4. Parthenogenetic Activation and In Vitro Culturing of Porcine Oocytes

Following culture maturation, cumulus cells were removed from the COCs using 0.1% hyaluronidase. Under a stereomicroscope, metaphase II (MII) oocytes exhibiting a uniform cytoplasm and first polar body extrusion were selected for chemical-induced parthenogenetic activation. Post-activation, embryos were immediately transferred to Porcine Zygote Medium-3 (PZM-3) and cultured at 38.5 °C in a humidified atmosphere containing 5% CO_2_. Cleavage and blastocyst formation rates were determined on Day 2 and Day 6, respectively.

### 4.5. Fluorescence Staining

Following culture maturation and cumulus cell denudation, selected oocytes underwent the following analytical procedures:

For ROS level detection, oocytes were incubated in PBS containing 10 μM of reactive oxygen species assay kit (dichlorofluorescein diacetate, DCFHDA), Beyotime Biotechnology, Shanghai, China) for 30 min in an incubator. Following washing with PBS containing 1% BSA, specimens were visualized and documented under a fluorescence microscope using standardized scanning parameters.

To determine mitochondrial content, according to manufacturer’s protocol, oocytes were incubated with MitoTracker Red CMXRos working solution (Beyotime, Shanghai, China) for 30 min in an incubator. After three successive washes with culture medium, specimens were examined and documented using standardized fluorescence microscopy settings.

For mitochondrial membrane potential assessment, following the protocol of the enhanced mitochondrial membrane potential assay kit with JC-1 (Beyotime, Shanghai, China), mature oocytes were incubated with JC-1 dye for 30 min in an incubator. After triple washing, specimens were analyzed under standardized fluorescence microscopy parameters. The red-to-green fluorescence intensity ratio was quantified using ImageJ software (v1.8.0).

Immunofluorescence staining was performed as follows: denuded oocytes (DOs) were fixed in 4% paraformaldehyde/PBS (30 min, room temperature (RT)), permeabilized in 1% Triton X-100/PBS (8–10 h), and blocked in 1% BSA/PBS (1–2 h, RT). Specimens were then incubated overnight (4 °C) with the following primary antibodies: lectin–FITC (1:100; Sigma-Aldrich, St. Louis, MO, USA), α-tubulin–FITC (1:200; Sigma-Aldrich), actin–Tracker Red (1:100; Beyotime, Shanghai, China), or anti-γ-H2A.X (1:100; ABclonal Technology, Wuhan, China). Following triple washing in PBST (5 min each), samples were incubated with goat anti-mouse IgG (H+L) (1:150; ABclonal Technology) for 1.5 h at RT. After washing, specimens were counterstained with Hoechst 33342 (10 μg/mL, 10 min, RT). Mounted samples were examined using a Leica TCS-SP8 laser scanning confocal microscope (Wetzlar, Germany).

Spindles were considered normal when they exhibited a typical barrel-shaped spindle apparatus or essentially symmetric. Spindles were considered abnormal (1) when they were small relative to the chromosomal metaphase plate, and/or (2) when they were “hairy”, with microtubules irregularly scattered throughout the cytoplasm. Chromosomes were considered normal when they aligned well with the equatorial plate or were only slightly scattered from the equatorial plate. When one or more chromosomes were situated completely outside the metaphase plate or when chromosomes were scattered throughout the spindle without displaying an equatorial plate, this was scored as unaligned [61,62].

To determine fluorescence intensity, images of control and treated oocytes were acquired using identical staining protocols and microscope parameters. Mean fluorescence intensity per unit area within regions of interest (ROI) was quantified using ImageJ software (NIH, Bethesda, MD, USA).

Blastocyst total cell count quantification was performed as follows: parthenogenetic blastocysts were fixed in 4% paraformaldehyde/PBS (30 min, RT), washed thrice with PBS–PVA, and then nuclear-stained with Hoechst 33342 (10 μg/mL, 10 min, RT). Following washing, mounted specimens were examined under an EVOS fluorescence microscope. Blastocyst total cells were quantified using ImageJ software.

### 4.6. ATP Analysis in Oocytes

Intracellular ATP levels were quantified using a commercial ATP Determination Kit (Beyotime Biotechnology, Shanghai, China) in accordance with the manufacturer’s specifications. After thorough PBS washing, twenty oocytes were pooled to constitute a single analytical sample, with three independent replicates (*n* = 3) per experimental group for ATP content determination.

### 4.7. Oocyte Micro-Reverse Transcription and Real-Time Quantitative PCR Analysis

Following culture termination, oocytes from each experimental group were harvested for transcriptional analysis. First-strand cDNA synthesis was performed using the SuperScriptTM II Reverse Transcription System (ThermoFisher Scientific, Waltham, MA, USA) in accordance with the manufacturer’s specifications. Quantitative PCR amplification was conducted utilizing 2×ChamQ Universal SYBR qPCR Master Mix (Vazyme, Nanjing, China). Gene expression levels were normalized to the constitutively expressed housekeeping gene *β-ACTIN* (Appendix A), and relative quantification was determined using the comparative threshold cycle (2^−ΔΔCt^) method. All experiments were performed independently in triplicate.

### 4.8. Statistical Analysis

All experimental data are expressed as mean ± standard error of the mean (SEM), with each experimental group comprising a minimum of three independent biological replicates. Statistical analyses were conducted using GraphPad Prism 7 software (GraphPad Software Inc., San Diego, CA, USA). Data were analyzed using a nested ANOVA in Section 2.1 and Section 2.2. Other results were analyzed by a one-way analysis of variance (ANOVA). Fluorescence intensity quantification was performed using NIH ImageJ software (National Institutes of Health, Bethesda, MD, USA). Differences were considered statistically significant when *p* < 0.05.

## 5. Conclusions

UA exhibits robust antioxidant properties and is capable of preserving mitochondrial function by modulating cellular and organismal autophagy levels. Our research has highlighted its beneficial impact on overall organismal health, the aging process, and various age-related health concerns. Nonetheless, the investigation of UA within the realm of reproductive science is relatively limited. In this study, we have demonstrated that the inclusion of UA in the IVM medium for porcine oocytes leads to several improvements. It enhances the oocytes’ antioxidant defenses, diminishes intracellular oxidative stress, reduces DNA damage resulting from oxidative injury, facilitates accurate spindle assembly and chromosomal alignment, regulates autophagy to reinstate mitochondrial functionality, and ensures the proper distribution of cortical granules. Collectively, these actions rectify the oxidative stress-induced meiotic abnormalities in oocytes and significantly bolster the developmental potential and quality of the ensuing parthenogenetic embryos.

## Figures and Tables

**Figure 1 ijms-26-03037-f001:**
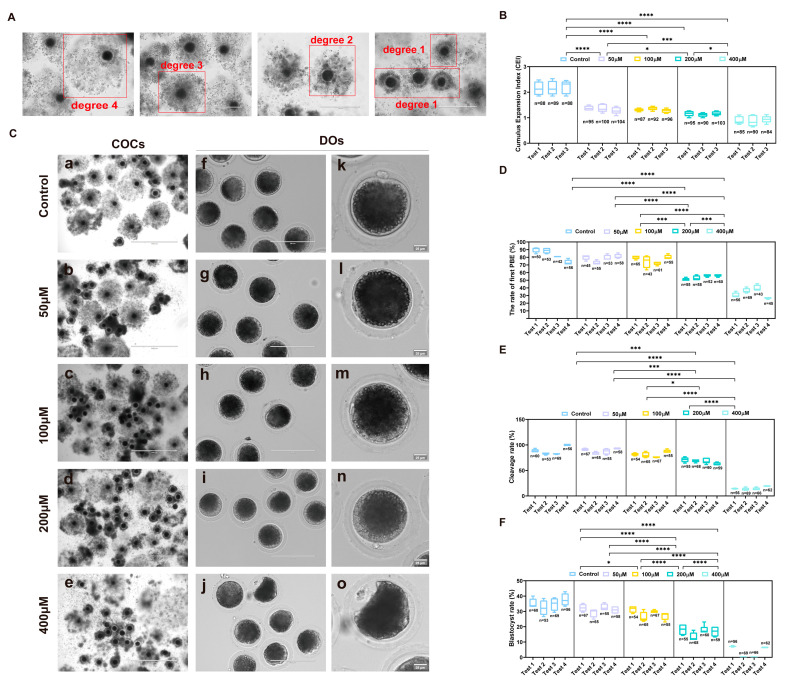
Effects of different concentrations of H_2_O_2_ on porcine oocyte maturation and parthenogenetic embryo developmental competence. (**A**) Illustrative examples demonstrating the cumulus expansion scoring system in porcine cumulus oocyte complexes (COCs). Scale bar, 400 µm. (**B**) Statistics of cumulus expansion after COCs were treated with different concentrations of H_2_O_2_. Control group *n* = 265, 50 μM group *n* = 299, 100 μM group *n* = 275, 200 μM group *n* = 288, 400 μM group *n* = 259. (**C**) Representative images of the control group and different concentrations H_2_O_2_-treated groups after in vitro maturation (IVM). Scale bar, 1000 µm (a–e); 200 µm (f–j); 25 µm (k–o). (**D**) The rate of the first polar body extrusion (PBE) was recorded in the control group and different concentrations of H_2_O_2_-treated groups. Control group *n* = 201, 50 μM group *n* = 211, 100 μM group *n* = 224, 200 μM group *n* = 225, 400 μM group *n* = 213. (**E**) The statistics of the cleavage rate recorded in the control group and different concentrations of H_2_O_2_-treated groups. Control group *n* = 238, 50 μM group *n* = 245, 100 μM group *n* = 241, 200 μM group *n* = 242, 400 μM group *n* = 253. (**F**) The rate of the blastocyst (BL) formation was recorded in the control group and different concentrations of H_2_O_2_-treated groups. Control group *n* = 238, 50 μM group *n* = 245, 100 μM group *n* = 241, 200 μM group *n* = 242, 400 μM group *n* = 253. Data in (**B**,**D**,**E**,**F**) are presented as the mean ± SEM of at least three independent experiments. * *p*  <  0.05, *** *p*  <  0.001, **** *p*  <  0.0001. The number of cells used for analysis was equal to the summation of cells used in each test.

**Figure 2 ijms-26-03037-f002:**
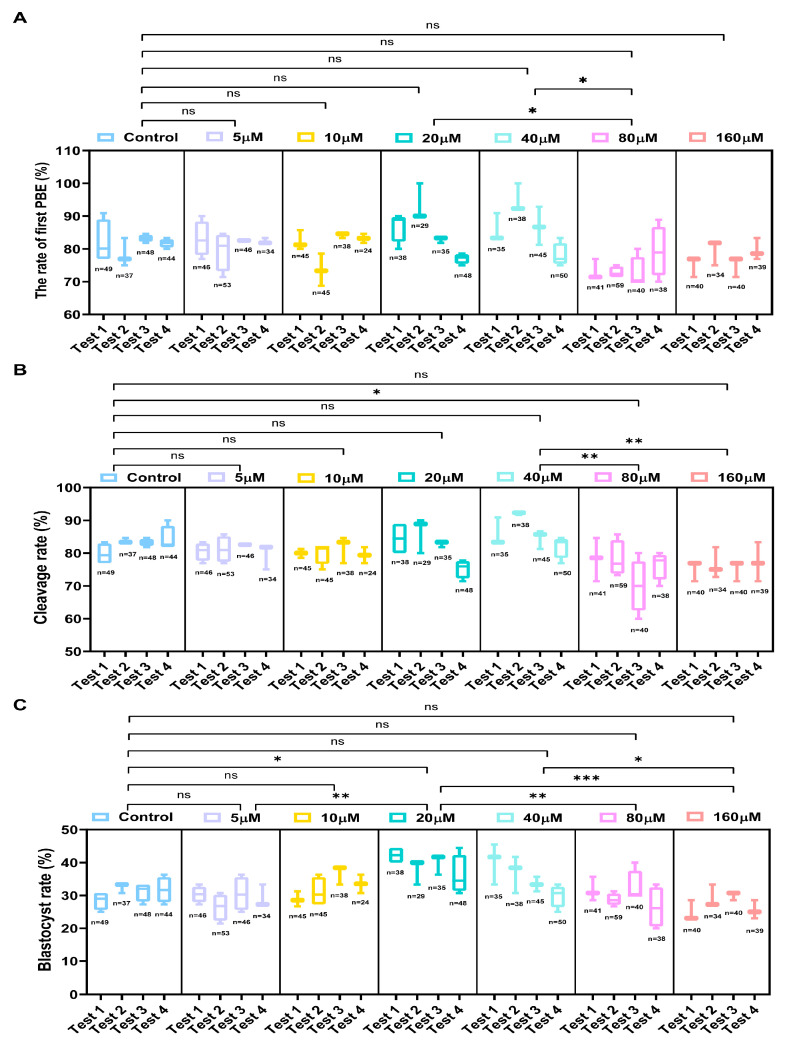
Effects of different doses of urolithin A (UA) supplement on porcine conventional oocyte IVM and parthenogenetic embryo developmental competence. (**A**) The rate of the first PBE of porcine conventional oocyte IVM was recorded in the control group and different concentrations of UA-treated groups. Control group *n* = 178, 5 μM group *n* = 179, 10 μM group *n* = 152, 20 μM group *n* = 150, 40 μM group *n* = 168, 80 μM group *n* = 178, 160 μM group *n* = 153. (**B**) The statistics of the cleavage rate recorded in the control group and different concentrations of UA-treated groups. Control group *n* = 178, 5 μM group *n* = 179, 10 μM group *n* = 152, 20 μM group *n* = 150, 40 μM group *n* = 168, 80 μM group *n* = 178, 160 μM group *n* = 153. (**C**) The rate of the BL formation was recorded in the control group and different concentrations of UA-treated groups. Control group *n* = 178, 5 μM group *n* = 179, 10 μM group *n* = 152, 20 μM group *n* = 150, 40 μM group *n* = 168, 80 μM group *n* = 178, 160 μM group *n* = 153. Data in (**A**–**C**) are presented as the mean ± SEM of at least three independent experiments. ns, no significance, * *p*  <  0.05, ** *p*  <  0.01, *** *p*  <  0.001. The number of cells used for analysis was equal to the summation of cells used in each test.

**Figure 3 ijms-26-03037-f003:**
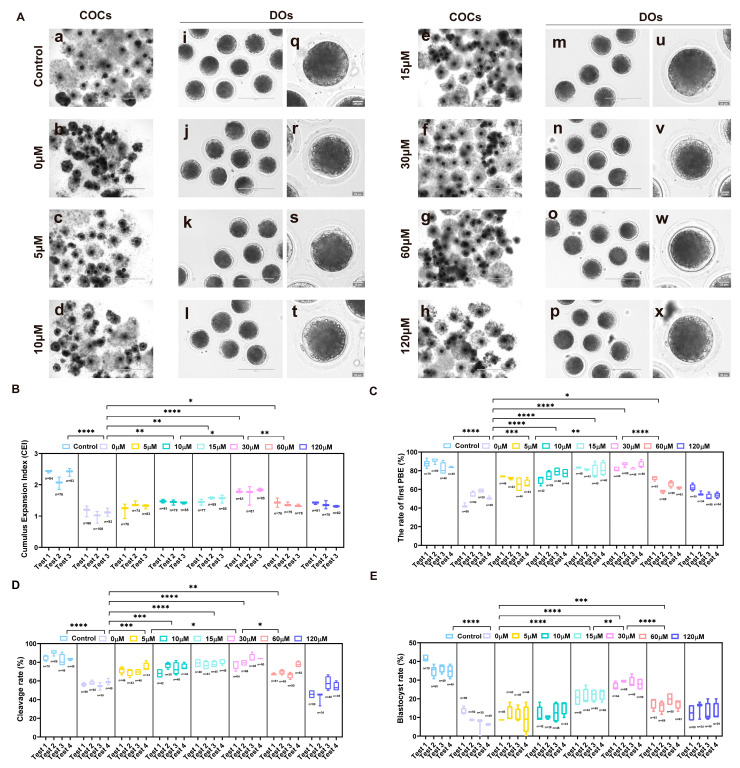
Effects of different concentrations of UA supplementation on porcine oocyte maturation and parthenogenetic embryo developmental competence. (**A**) Representative images of the control group and different concentrations UA supplementation after in vitro maturation (IVM). Scale bar, 200 µm (a–h,m–t); 25 µm (i–l,u–x). (**B**) Statistics of cumulus expansion after COCs were treated with different concentrations of UA. Control group *n* = 253, 0 μM group *n* = 285, 5 μM group *n* = 233, 10 μM group *n* = 245, 15 μM group *n* = 251, 30 μM group *n* = 233, 60 μM group *n* = 233, 120 μM group *n* = 223. (**C**) The rate of the first PBE was recorded in the control group and different concentrations of UA supplementation groups. Control group *n* = 246, 0 μM group *n* = 274, 5 μM group *n* = 179, 10 μM group *n* = 173, 15 μM group *n* = 184, 30 μM group *n* = 281, 60 μM group *n* = 243, 120 μM group *n* = 194. (**D**) The statistics of the cleavage rate recorded in the control group and different concentrations of UA supplementation groups. Control group *n* = 246, 0 μM group *n* = 274, 5 μM group *n* = 179, 10 μM group *n* = 173, 15 μM group *n* = 184, 30 μM group *n* = 227, 60 μM group *n* = 243, 120 μM group *n* = 163. (**E**) The rate of the blastocyst formation was recorded in the control group and different concentrations of UA supplementation groups. Control group *n* = 246, 0 μM group *n* = 274, 5 μM group *n* = 179, 10 μM group *n* = 173, 15 μM group *n* = 184, 30 μM group *n* = 227, 60 μM group *n* = 243, 120 μM group *n* = 163. Data in (**B**–**E**) are presented as the mean ± SEM of at least three independent experiments. * *p*  <  0.05, ** *p*  <  0.01, *** *p*  <  0.001, **** *p*  <  0.0001. The number of cells used for analysis was equal to the summation of cells used in each test.

**Figure 4 ijms-26-03037-f004:**
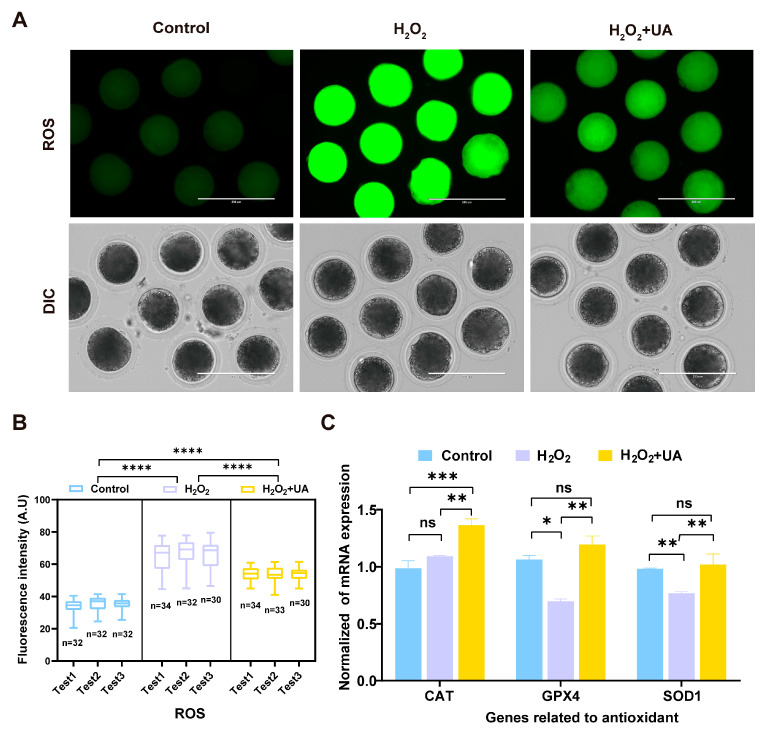
Antioxidant effect of UA on artificially induced oxidative stress-damaged porcine oocytes during IVM. (**A**) Representative images of the control group, H_2_O_2_-treated, and UA-supplemented oocytes stained with DCFHDA. Scale bar, 200 µm. (**B**) The fluorescence intensity of ROS signals was measured in the control group, H_2_O_2_-treated, and UA-supplemented oocytes. Control group *n* = 96, H_2_O_2_-treated group *n* = 96, H_2_O_2_+UA group *n* = 97. (**C**) Relative expression of genes related to antioxidant effects. Data are the mean ± SEM values. Each experiment was independently repeated at least three times. Data in (**B**,**C**) are presented as the mean ± SEM of at least three independent experiments. ns, no significance, * *p*  <  0.05, ** *p*  <  0.01, *** *p*  <  0.001, **** *p*  <  0.0001.

**Figure 5 ijms-26-03037-f005:**
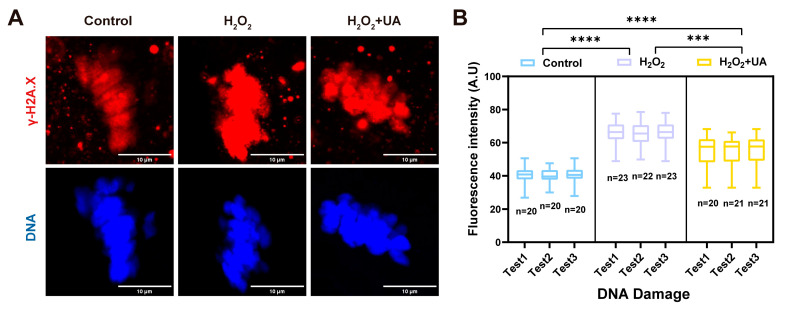
Effect of UA supplementation on DNA damage in artificially induced oxidative stress-damaged Porcine oocytes during IVM. (**A**) Representative images of DNA damage stained with the γ-H2AX antibody in the control group, H_2_O_2_-treated and UA-supplemented oocytes. Scale bar, 10 μm. (**B**) The fluorescence intensity of γ-H2A.X signals were measured in the control group, H_2_O_2_-treated and UA-supplemented oocytes. Control group *n* = 60, H_2_O_2_-treated group *n* = 68, H_2_O_2_+UA group *n* = 62. Data in (**B**) are presented as the mean ± SEM of at least three independent experiments. *** *p*  <  0.0001; **** *p*  <  0.0001.

**Figure 6 ijms-26-03037-f006:**
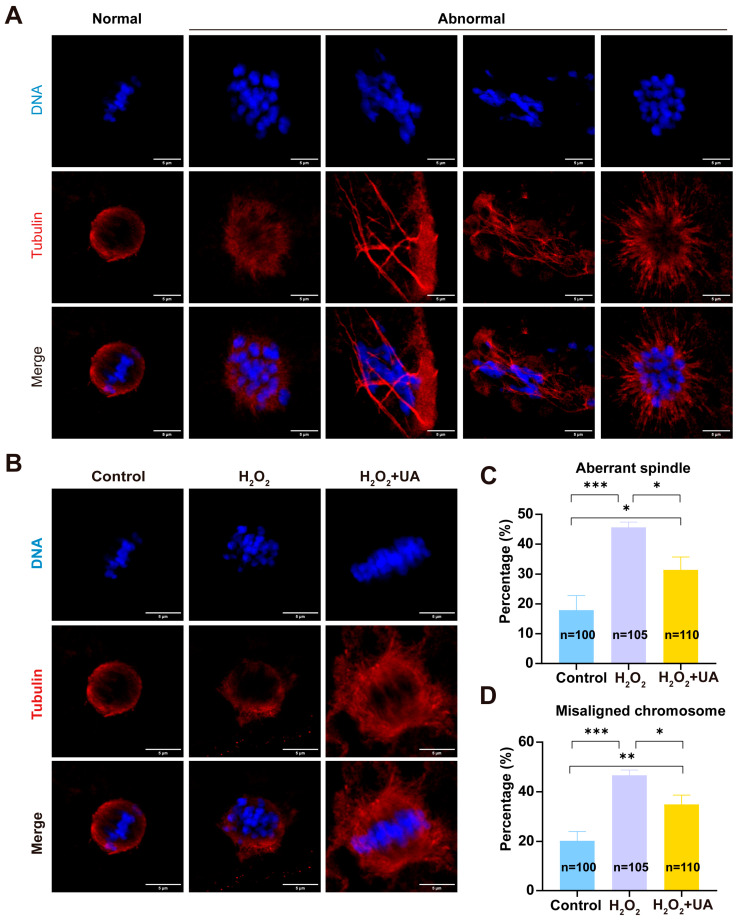
Effect of UA Supplementation on spindle and chromosome structure in artificially induced oxidative stress-damaged porcine oocytes during IVM. (**A**) Representative images of normal and abnormal spindle morphology and chromosome alignment at metaphase II. Scale bar, 5 μm. (**B**) Representative images of the spindle morphology and chromosome alignment at metaphase II in the control group, H_2_O_2_-treated, and UA-supplemented oocytes. Scale bar, 5 μm. (**C**) The rate of aberrant spindles at metaphase II was recorded in the control group, H_2_O_2_-treated, and UA-supplemented oocytes. Control group *n* = 100, H_2_O_2_-treated group *n* = 105, H_2_O_2_+UA group *n* = 110. (**D**) The rate of misaligned chromosomes at metaphase II was recorded in the control group, H_2_O_2_-treated, and UA-supplemented oocytes. Control group *n* = 100, H_2_O_2_-treated group *n* = 105, H_2_O_2_+UA *n* = 110 group. Data in (**C**,**D**) are presented as the mean ± SEM of at least three independent experiments. * *p*  <  0.05, ** *p*  <  0.01, *** *p*  <  0.001.

**Figure 7 ijms-26-03037-f007:**
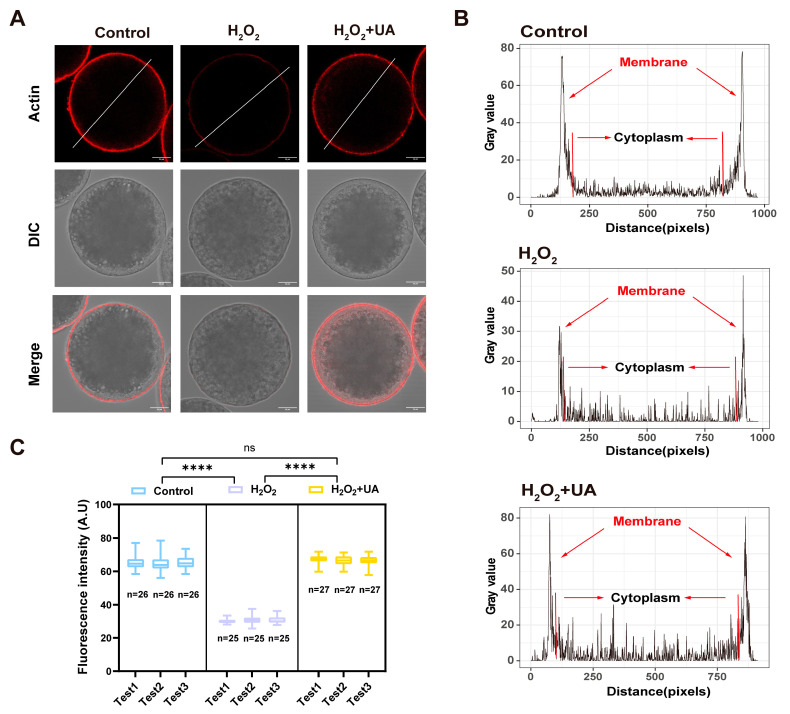
Effect of UA supplementation on actin cytoskeleton dynamics in artificially induced oxidative stress-damaged porcine oocytes during IVM. (**A**) Fluorescence images of actin cytoskeleton signals at metaphase II in the control group, H_2_O_2_-treated, and UA-supplemented oocytes. Scale bar, 20 μm. (**B**) The fluorescence intensity profiling of actin in the control group, H_2_O_2_-treated, and UA-supplemented oocytes. (**C**) The fluorescence intensity of actin on the plasma membrane was measured in the control group, H_2_O_2_-treated, and UA-supplemented oocytes. Control group *n* = 78, H_2_O_2_-treated group *n* = 75 and H_2_O_2_+UA group *n* = 81. Data in (**C**) are presented as the mean ± SEM of at least three independent experiments. ns, no significance, **** *p*  <  0.001.

**Figure 8 ijms-26-03037-f008:**
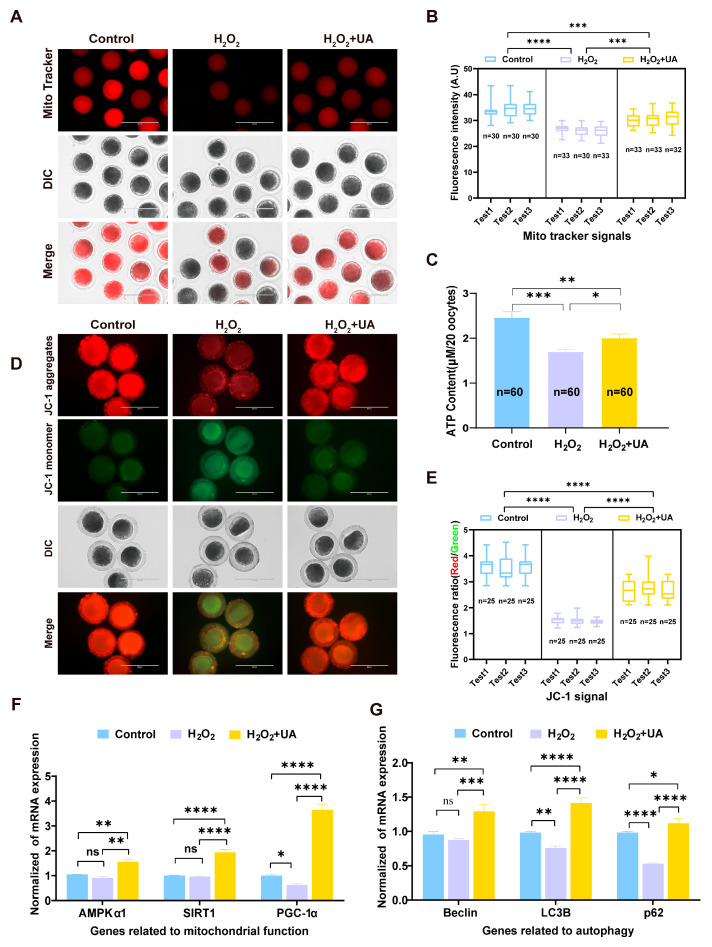
Effect of UA supplementation on mitochondrial function in artificially induced oxidative stress-damaged porcine oocytes during IVM. (**A**) Representative images of mitochondria in the control group, H_2_O_2_-treated, and UA-supplemented oocytes. Scale bar, 200 µm. (**B**) The fluorescence intensity of MitoTracker signals was recorded in the control group, H_2_O_2_-treated, and UA-supplemented oocytes. Control group *n* = 90, H_2_O_2_-treated group *n* = 96, H_2_O_2_+UA group *n* = 98. (**C**) ATP levels were measured in the control group, H_2_O_2_-treated, and UA-supplemented oocytes. Control group *n* = 60, H_2_O_2_-treated group *n* = 60, H_2_O_2_+UA group *n* = 60. (**D**) Mitochondrial membrane potential was detected by JC-1 staining in the control group (*n* = 25), H_2_O_2_-treated (*n* = 25), and UA-supplemented (*n* = 24) oocytes. Scale bar, 200 μm. (**E**) The ratio of red to green fluorescence intensity was calculated in the control group, H_2_O_2_-treated, and UA-supplemented oocytes. Control group *n* = 75, H_2_O_2_-treated group *n* = 75, H_2_O_2_+UA group *n* = 75. (**F**) Relative expression of genes related to mitochondrial function. (**G**) Relative expression of genes related to autophagy. Data in (**B**,**C**,**E**–**G**) are presented as the mean ± SEM of at least three independent experiments. ns, no significance, * *p*  <  0.05, ** *p*  <  0.01, *** *p*  <  0.001, **** *p*  <  0.0001.

**Figure 9 ijms-26-03037-f009:**
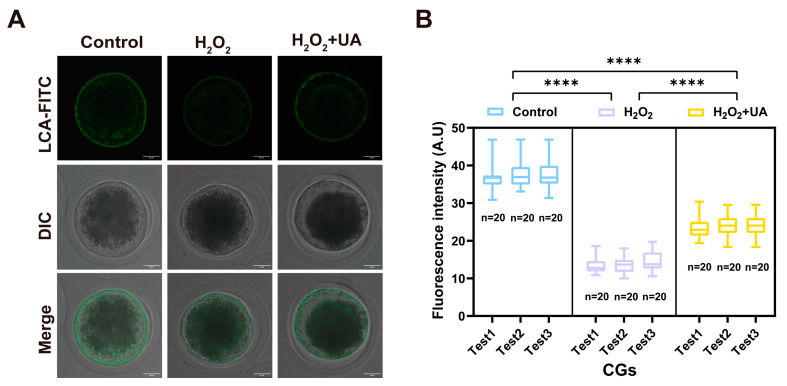
Effect of UA supplementation on the dynamics of CGs in artificially induced oxidative stress-damaged porcine oocytes during IVM. (**A**) Representative images of CG distribution in the control group, H_2_O_2_-treated, and UA-supplemented oocytes. Scale bar, 25 µm. (**B**) The fluorescence intensity of CGs signals was measured in the control group, H_2_O_2_-treated, and UA-supplemented oocytes. Control group *n* = 60, H_2_O_2_-treated group *n* = 60, H_2_O_2_+UA group *n* = 60. Data in (**B**) are presented as the mean percentage (mean ± SEM) of at least three independent experiments. **** *p*  <  0.0001.

**Figure 10 ijms-26-03037-f010:**
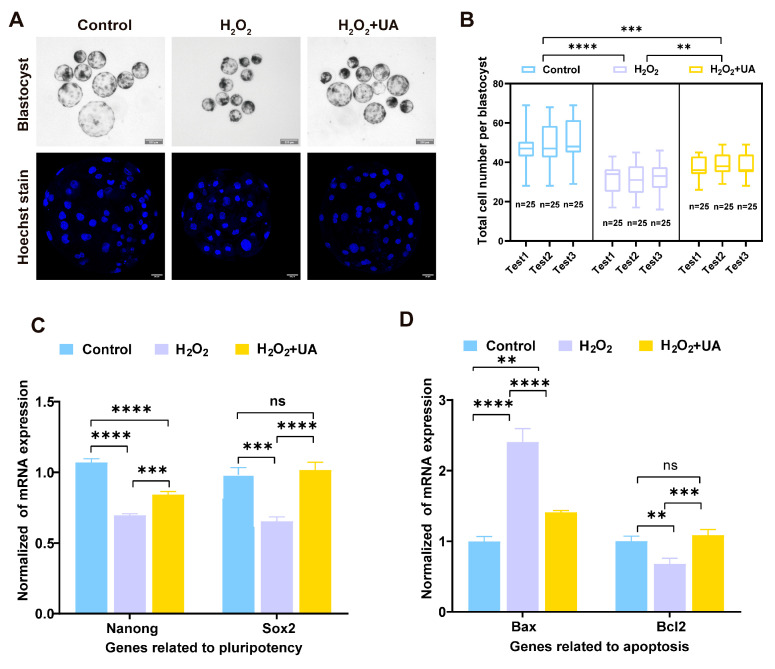
Effect of UA supplementation on the developmental quality of parthenogenetic embryos. (**A**) Representative images of blastocysts (scale bar, 200 µm) and Hoechst stain in blastocysts (scale bar, 25 µm) from the control group (*n* = 25), H_2_O_2_-treated (*n* = 25), and UA-supplemented (*n* = 25). (**B**) Statistics of total cell number per blastocyst in the control group, H_2_O_2_-treated, and UA-supplemented oocytes. Control group *n* = 75, H_2_O_2_-treated group *n* = 75, H_2_O_2_+UA group *n* = 75. (**C**) Relative expression of genes related to pluripotency. Data are the mean ± SEM values. (**D**) Relative expression of genes related to apoptosis. Data are the mean ± SEM values. Data in (**B**–**D**) are presented as the mean ± SEM of at least three independent experiments. ns, no significance, ** *p*  <  0.01, *** *p*  <  0.001, **** *p*  <  0.0001.

## Data Availability

The original contributions presented in this study are included in the article/Appendix A. Further inquiries can be directed to the corresponding authors.

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
