# Peer review of "Urolithin A Protects Porcine Oocytes from Artificially Induced Oxidative Stress Damage to Enhance Oocyte Maturation and Subsequent Embryo Development"

_ijms, 2025, doi:10.3390/ijms26073037_

Round 1
Reviewer 1 Report
Comments and Suggestions for Authors
In this manuscript, the authors explore the role of urolithin A as an antioxidant agent for improving oocyte quality and, consequently, the success of in vitro fertilisation techniques. The paper is well-written and appears to be scientifically sound for the most part. However, I have a few suggestions that could enhance the manuscript.
- In particular, I have some concerns regarding the choice to perform parthenogenesis to assess embryonic development rather than the more physiological fertilisation with capacitated spermatozoa. Is there a specific reason why the authors made this choice? If so, it should be justified and included in the manuscript. Otherwise, it would be better to follow the embryonic development after fertilization.
- Among the various parameters for assessing oocyte quality, the authors chose to use tubulin to evaluate the formation of the meiotic spindle as an indicator of cytoskeletal state. They also assessed the distribution of cortical granules in the cortex of oocytes, which, even in mammals (as in many other organisms), is mediated by microfilaments. This is demonstrated by numerous previous works in which agents such as cytochalasin B or jasplakinolide alter their movement or even their exocytosis at fertilisation. It would, therefore, be interesting to evaluate the state of the actin cytoskeleton in oocytes subjected to oxidative stress and subsequent treatment with urolithin A.
Minor points:
- The unit is missing on the y-axis of the histograms in the figures where fluorescence intensity is evaluated. Is it the standard AU or another?
- On the y-axis of the histograms of Figure 5, the word "percentage" is misspelt.
- Line 276 H202 contains an extra H.
Author Response
For research article
|
Response to Reviewer 1 Comments
|
||
|
1. Summary |
|
|
|
Thank you very much for taking the time to review this manuscript. We have revised our manuscripts point by point according to these comments and suggestions. Please find the detailed responses below and the corresponding revisions/corrections highlighted/in track changes in the re-submitted files. We wish that our revised manuscript will be qualified enough to be accepted by the journal.
|
||
|
2. Questions for General Evaluation |
Reviewer’s Evaluation |
Response and Revisions |
|
Does the introduction provide sufficient background and include all relevant references? |
Yes/Can be improved/Must be improved/Not applicable |
|
|
Are all the cited references relevant to the research? |
Yes/Can be improved/Must be improved/Not applicable |
|
|
Is the research design appropriate? |
Yes/Can be improved/Must be improved/Not applicable |
|
|
Are the methods adequately described? |
Yes/Can be improved/Must be improved/Not applicable |
|
|
Are the results clearly presented? |
Yes/Can be improved/Must be improved/Not applicable |
|
|
Are the conclusions supported by the results? |
Yes/Can be improved/Must be improved/Not applicable |
|
|
3. Point-by-point response to Comments and Suggestions for Authors |
||
|
Comments 1: In particular, I have some concerns regarding the choice to perform parthenogenesis to assess embryonic development rather than the more physiological fertilisation with capacitated spermatozoa. Is there a specific reason why the authors made this choice? If so, it should be justified and included in the manuscript. Otherwise, it would be better to follow the embryonic development after fertilization.
|
||
|
Response 1: |
||
|
Thank you for your valuable comments. Compared with other mammals, the polyspermic fertilization rate of pigs is very high, the polyspermic fertilization rate in vivo can reach 30%-40%, and the polyspermic fertilization rate in vitro can exceed 65% and lead to the formation of polyploid embryos, which seriously limits the application of pig in vitro fertilization (IVF) technology and in vitro embryo production. IVF tests were also conducted during our study, but due to the serious problem of polyspermic fertilization, the subsequent development of IVF embryos was seriously hindered, and fewer normal development of IVF embryos were obtained. The results of IVF test showed that the addition of UA slightly increased the development of porcine IVF embryos under oxidative stress (Control 8.33 ± 0.21%, n=305 vs. H2O2 2.90 ± 1.01%, n=288, P<0.01; Control 8.33 ± 0.21%, n=305 vs.H2O2+UA 3.93 ± 0.56%, n=288, P <0.05; H2O2 2.90 ± 1.01%, n=288 vs.H2O2+UA 3.93 ± 0.56%, n=288, P >0.05, Table S6). Relevant statistics were supplemented in Table S6 of supplementary materials and described in line 176-178 of the revised manuscript. Representative images of UA Supplement on H2O2-treated Porcine In Vitro Fertilization have also added in Figure S1 of supplementary materials. Therefore, in our study we chose parthenogenesis activation to instead of in vitro fertilization for follow-up tests.
Comments 2: Among the various parameters for assessing oocyte quality, the authors chose to use tubulin to evaluate the formation of the meiotic spindle as an indicator of cytoskeletal state. They also assessed the distribution of cortical granules in the cortex of oocytes, which, even in mammals (as in many other organisms), is mediated by microfilaments. This is demonstrated by numerous previous works in which agents such as cytochalasin B or jasplakinolide alter their movement or even their exocytosis at fertilisation. It would, therefore, be interesting to evaluate the state of the actin cytoskeleton in oocytes subjected to oxidative stress and subsequent treatment with urolithin A. |
||
|
Response 2: Thank you for your valuable comments. As you mentioned, actin cytoskeleton plays a key role in nuclear positioning, spindle migration and anchoring, and polar body extrusion to promote meiotic progression in mammalian oocytes. We therefore used phalloidin to visualize the changes of actin polymerization in H2O2-treated oocytes exposed to UA. Our results showed that UA maintains the actin cytoskeleton during oxidative stress to protect the oocyte integrity. The specific results were added in revised manuscript “2.6 UA supplementation rescues the actin polymerization in H2O2-Treated Porcine Oocytes”, in line 227-245.
Comments 3: The unit is missing on the y-axis of the histograms in the figures where fluorescence intensity is evaluated. Is it the standard AU or another? Response3: Thank you for your kind suggestion. The unit of the fluorescence intensity statistical graph in the figures is A.U. We have corrected this mistake in our manuscript, such as Figure 3B, Figure 4B, Figure 6C and Figure 8B.
Comments 4: On the y-axis of the histograms of Figure 5, the word "percentage" is misspelt. Response4: Thank you for your kind suggestion. We have corrected the incorrect spelling, the word "percentge" in the histograms of Figure 5 has changed into “percentage”, in line 225.
Comments 5: Line 276 H202 contains an extra H. We were really sorry for our careless mistake. Thank you for pointing this out. We have corrected the mistake, the extra “H” has eliminated.
|
||

Reviewer 2 Report
Comments and Suggestions for Authors
In this study, the authors analysed if antioxidant Urolithin A (UA) improves the quality of porcine oocytes obtained by in vitro maturation (IVM). The authors induced oxidative stress by 200uM of H2O2, then supplemented the medium with 30uM of UA and analysed a wide range of oxygenation stress-related parameters: (i) level of reactive oxidant species (ROS) in oocytes; (ii) DNA damage by gH2AX staining, (iii) spindle morphology and chromosome alignment at the metaphase II stage; (iv) mitochondrial content and membrane potential; (v) ATP content, (vi) expression of oxygen-protective genes, mitochondrial function-related genes, autophagy-related genes; (vi) cytoplasmic maturation competence by cortical granules distribution, and finally (vii) development competence of parthenogenetically activated oocytes to the blastocyst stage.
Overall, the study is well done, and all the reported parameters are relevant. However, the in this study the authors did not explore the mechanism by which UA protects oocytes during IVM in this work. Some important issues need to be addressed before the study can be published.
Major points:
- The study demonstrates the protective properties of UA on porcine oocytes only after artificially induced oxidative stress (exposure to H2O2). As noted in the introduction, the IVM itself induces oxidative stress in oocytes. Therefore, it is crucial to determine if UA supplementation would improve oocyte quality derived from the regular IVM procedure. Without this verification, the study remains incomplete, as it is unclear if UA would be beneficial in improving porcine oocyte quality in assisted reproductive technologies or if it is only relevant after specific oxidative damage induction.
- The most clear readout of the oocyte quality is their developmental competence. The authors looked only into the average number of cells in blastocysts at day 6 post artificial activation. It is of relevance to access oocyte quality by other parameters, like blastocyst morphology with a presence of inner cell mass and trophectoderm layer; incidence of apoptosis; mitochondrial function or DNA damage markers.
- The study conclusions are based on statistical comparisons between the groups. However, the statistical analysis is not well described. Please provide the name of the test each time a p-value is mentioned. Note that your data has a hierarchical structure, so the tests need to account for this. The data would be better understood if you provide either dots corresponding to the individual measurements or use box-and-whisker plots to better illustrate data variability. Additionally, provide the number of objects included in each group on each plot in your manuscript.
Minor points:
- Better describe the components used for analyzing different properties: explain what DCFHDA and gH2AX are; describe why red/green ratio is measured for the JC-1 probe, explain how the genes were picked for the qPCR, etc.. Also, provide references to publications demonstrating the relevance of the studied parameters to the oocyte/blastocyst quality.
- Fig.5: clarify the parameters for counting aberrant spindles and misaligned chromosomes.
- In some figures (e.g., Fig. 1D), the axis titles are too general. Make all axis titles descriptive so readers do not need to search the figure legend for explanations.
- Improve figure quality: In Fig. 7A, LCA-FITC is not visible; in Fig. 8A, Hoechst is barely seen.
Check spelling and grammar: For example, there is a spelling mistake in the title and in the axis title of Fig. 2E.
Author Response
For research article
|
Response to Reviewer 2 Comments
|
||
|
1. Summary |
|
|
|
Thank you very much for taking the time to review this manuscript. We have revised our manuscripts point by point according to these comments and suggestions. Please find the detailed responses below and the corresponding revisions/corrections highlighted/in track changes in the re-submitted files. We wish that our revised manuscript will be qualified enough to be accepted by the journal.
|
||
|
2. Questions for General Evaluation |
Reviewer’s Evaluation |
Response and Revisions |
|
Does the introduction provide sufficient background and include all relevant references? |
Yes/Can be improved/Must be improved/Not applicable |
|
|
Are all the cited references relevant to the research? |
Yes/Can be improved/Must be improved/Not applicable |
|
|
Is the research design appropriate? |
Yes/Can be improved/Must be improved/Not applicable |
|
|
Are the methods adequately described? |
Yes/Can be improved/Must be improved/Not applicable |
|
|
Are the results clearly presented? |
Yes/Can be improved/Must be improved/Not applicable |
|
|
Are the conclusions supported by the results? |
Yes/Can be improved/Must be improved/Not applicable |
|
|
3. Point-by-point response to Comments and Suggestions for Authors
|
||
|
Comments 1: The study demonstrates the protective properties of UA on porcine oocytes only after artificially induced oxidative stress (exposure to H2O2). As noted in the introduction, the IVM itself induces oxidative stress in oocytes. Therefore, it is crucial to determine if UA supplementation would improve oocyte quality derived from the regular IVM procedure. Without this verification, the study remains incomplete, as it is unclear if UA would be beneficial in improving porcine oocyte quality in assisted reproductive technologies or if it is only relevant after specific oxidative damage induction. |
||
|
Response 1: |
||
|
Thank you for your comments. We have tested the effect during porcine oocyte IVM, but our purpose of this study is to explore the effect of UA on porcine oocytes under oxidative stress model. In the process of in vitro culture, we could not conduct a control test under physiological oxygen conditions (low oxygen conditions) due to conditions limitations, so we only tested the effect of adding UA to conventional IVM (under normal oxygen conditions) on porcine oocyte quality and parthenogenetic embryo development potential. Our results showed that addition of UA could promote the IVM rates and subsequent embryonic development of porcine oocytes. The relevant results have already mentioned in the revised manuscript, in lines 143-144, and the results have attached in supplementary material Table S7.
Comments 2: The most clear readout of the oocyte quality is their developmental competence. The authors looked only into the average number of cells in blastocysts at day 6 post artificial activation. It is of relevance to access oocyte quality by other parameters, like blastocyst morphology with a presence of inner cell mass and trophectoderm layer; incidence of apoptosis; mitochondrial function or DNA damage markers. |
||
|
Response 2: Thank you for your comments. We have added the testing of the mRNA levels of genes related to the developmental potential of parthenogenetic blastocysts (Nanog and Sox2) and apoptosis (Bax and Bcl2) in order to better proof UA supplement can effectively mitigate the adverse effects of oxidative stress on porcine oocytes and their developmental competence. These results had added in the revised manuscript (line 326-339).
Comments 3: The study conclusions are based on statistical comparisons between the groups. However, the statistical analysis is not well described. Please provide the name of the test each time a p-value is mentioned. Note that your data has a hierarchical structure, so the tests need to account for this. The data would be better understood if you provide either dots corresponding to the individual measurements or use box-and-whisker plots to better illustrate data variability. Additionally, provide the number of objects included in each group on each plot in your manuscript.
Response3: Thank you for your comments. We were really sorry for our careless mistake. The test in our study is one-way analysis of variance (ANOVA), as we have mentioned in “4.8 Statistical Analysis” (line 547-548). We have changed the fluorescence intensity statistical graph into scatter plot to better illustrate data variability, such as Figure 3B, Figure 4B, Figure 6B, Figure 7B and E, Figure 8B and Figure 9B. We have added the number of objects included in each group on each plot in our manuscript, changes have been highlighted in yellow.
Comments 4: Better describe the components used for analyzing different properties: explain what DCFHDA and gH2AX are; describe why red/green ratio is measured for the JC-1 probe, explain how the genes were picked for the qPCR, etc.. Also, provide references to publications demonstrating the relevance of the studied parameters to the oocyte/blastocyst quality. Response4: Thank you for your kind suggestion. (1) Dichlorofluorescein diacetate (DCFHDA) is the oxidationsensitive fluorescent probe to detects intracellular reactive oxygen species levels, we have added the explanation in line 186-187. (2) γ-H2A.X is the key regulator of DNA damage, we have added the explanation in line 198. (3) In JC-1 fluorescent test, yields green or red fluorescence, to distinguish cells with high or low mitochondria potential. The mitochondria with a high membrane potential fluoresced red, whereas those with a low membrane potential fluoresced green. We have added the explanation in line 264-266. (4) SOD1, CAT, and GPX4 are antioxidant-related genes,we tested these genes in order to further proof whether UA supplementation can restored antioxidant capacity of in H2O2-treated porcine oocytes following oxidative stress. Mitochondrial function related genes AMPKα1, SIRT1 and PGC-1α were tested to explore the effect on mitochondrial function of UA supplementation. Autophagy helps to remove damaged mitochondria and restore mitochondrial function, so we tested the expression of autophagy related genes (Beclin, LC3B and p62) in order to better explore the potential mechanisms of UA supplementation to enhance mitochondrial function. The mRNA levels of genes related to the developmental potential of parthenogenetic blastocysts (Nanog and Sox2) and apoptosis (Bax and Bcl2) were added to measure, in order to better proof UA supplementation effect in enhancing developmental quality of parthenogenetic embryos derived from H2O2-treated porcine oocytes. (5) Relative references to publications demonstrating the relevance of the studied parameters to the oocyte/blastocyst quality have already added in line 57.
Comments 5: Fig.5: clarify the parameters for counting aberrant spindles and misaligned chromosomes. Response5: Thank you for your kind suggestion. The parameters for counting aberrant spindles and misaligned chromosomes have added in Figure 5A.
Comments 6: In some figures (e.g., Fig. 1D), the axis titles are too general. Make all axis titles descriptive so readers do not need to search the figure legend for explanations. Response6: Thank you for your kind suggestion. We have added descriptions to the title of the statistical graph for a better understanding of our result, such as Figure 1D, Figure 2C, Figure 3C, Figure 4B, Figure 6F andG, Figure 8B and Figure 9C and D.
Comments 7: Improve figure quality: In Fig. 7A, LCA-FITC is not visible; in Fig. 8A, Hoechst is barely seen. Response7: Thank you for your kind suggestion. We have tried our best to obtain high quality fluorescence images by a laser scanning confocal microscope, the images in Fig. 7A and Fig. 8A may have become blurred due to the compression of the file when uploading the manuscript. We rearranged the PDF of all the images and uploaded them together with the revised manuscript.
|
||
|
4. Response to Comments on the Quality of English Language |
||
|
Point 1: Check spelling and grammar: For example, there is a spelling mistake in the title and in the axis title of Fig. 2E. |
||
|
Response 1: |
||
Thank you for your kind suggestion. The spelling mistake of “blastocyst” in the title and in the axis title of Fig. 2E has corrected.

Round 2
Reviewer 1 Report
Comments and Suggestions for Authors
The authors have addressed all the points I raised by performing the requested experiments.
Author Response
Dear Reviewer, We would like to extend our heartfelt gratitude for your professional review of our manuscript and the constructive comments with insightful suggestions, which have significantly enhanced the academic rigor and clarity of our work.
Reviewer 2 Report
Comments and Suggestions for Authors
Comments 1: The study demonstrates the protective properties of UA on porcine oocytes only after artificially induced oxidative stress (exposure to H2O2). As noted in the introduction, the IVM itself induces oxidative stress in oocytes. Therefore, it is crucial to determine if UA supplementation would improve oocyte quality derived from the regular IVM procedure. Without this verification, the study remains incomplete, as it is unclear if UA would be beneficial in improving porcine oocyte quality in assisted reproductive technologies or if it is only relevant after specific oxidative damage induction.
Response 1:
Thank you for your comments. We have tested the effect during porcine oocyte IVM, but our purpose of this study is to explore the effect of UA on porcine oocytes under oxidative stress model. In the process of in vitro culture, we could not conduct a control test under physiological oxygen conditions (low oxygen conditions) due to conditions limitations, so we only tested the effect of adding UA to conventional IVM (under normal oxygen conditions) on porcine oocyte quality and parthenogenetic embryo development potential. Our results showed that addition of UA could promote the IVM rates and subsequent embryonic development of porcine oocytes. The relevant results have already mentioned in the revised manuscript, in lines 143-144, and the results have attached in supplementary material Table S7.
Thank you for your answer.
On lines 143-144 you state:
“Our results showed that addition of UA could promote the conventional IVM rates and subsequent embryonic development of porcine oocytes(Table S7)”.
This text does not accurately convey the findings presented in Table S7. Table S7 shows that the addition of 20 µM UA to the IVM medium does not have an effect on PBE and embryo cleavage but increases the blastocyst rate from ≈31% to ≈36%. It should be noted, though, that the beneficial effect of this addition cannot be verified without analyzing blastocyst quality, similar to the approach in Figure 9. Other concentrations of UA either do not have any effects or reduce fitness.
The data presented in Table S7 is of significant interest, as it shows the potential of UA treatment to improve porcine blastocyst maturation and the scope of the expected effect. Therefore, this data should be incorporated into the main text of the article, either as a table in its current form or as a graph, and the results should be discussed in the Discussion section.
The work in the present form demonstrates the protective effect of the UA only after oxidative stress induced by H2O2. Therefore, the title needs to be revised to clearly convey this, for example: “Urolithin A Protects Porcine Oocytes from Artificially Induced Oxidative Stress Damage, Enhancing Oocyte Maturation and Subsequent Embryo Development”.
Note that the Table S7 has a mistake, the heading of the first column should be Concentration of UA , not of H2O2.
Comments 2: The most clear readout of the oocyte quality is their developmental competence. The authors looked only into the average number of cells in blastocysts at day 6 post artificial activation. It is of relevance to access oocyte quality by other parameters, like blastocyst morphology with a presence of inner cell mass and trophectoderm layer; incidence of apoptosis; mitochondrial function or DNA damage markers.
Response 2:
Thank you for your comments. We have added the testing of the mRNA levels of genes related to the developmental potential of parthenogenetic blastocysts (Nanog and Sox2) and apoptosis (Bax and Bcl2) in order to better proof UA supplement can effectively mitigate the adverse effects of oxidative stress on porcine oocytes and their developmental competence. These results had added in the revised manuscript (line 326-339).
Thank you for your answer.
Although it's regrettable that the authors cannot offer a more detailed analysis of blastocysts under UA protection except for the apoptotic markers, the challenges are understandable. I have no further comments here.
Comments 3: The study conclusions are based on statistical comparisons between the groups. However, the statistical analysis is not well described. Please provide the name of the test each time a p-value is mentioned. Note that your data has a hierarchical structure, so the tests need to account for this. The data would be better understood if you provide either dots corresponding to the individual measurements or use box-and-whisker plots to better illustrate data variability. Additionally, provide the number of objects included in each group on each plot in your manuscript.
Response3:
Thank you for your comments. We were really sorry for our careless mistake. The test in our study is one-way analysis of variance (ANOVA), as we have mentioned in “4.8 Statistical Analysis” (line 547-548).
Thank you for your answer.
In lines 547-548 you state: “…one-way ANOVA applied where appropriate”; but it is not clear which statistical test was used when it was not appropriate?
Your data has a hierarchical structure, so Nested ANOVA would be more appropriate to use. For example, in the comparison shown in Figure 3B, the DCFHDA fluorescence intensities are the dependent variable that should be analyzed in relation to the main factor (treatment type) and also the nested factor (independent experiments). Please note that in some cases (for example, Figure 7B), you might lose statistical significance; please adjust the text accordingly if that is the case. It would be particularly interesting to see if the significance criteria are met when re-analyzing the data for blastocyst IVM in the presence of 20 uMUA, presented in Figure S7.
We have changed the fluorescence intensity statistical graph into scatter plot to better illustrate data variability, such as Figure 3B, Figure 4B, Figure 6B, Figure 7B and E, Figure 8B and Figure 9B.
Thank you for your answer.
Please make the lines corresponding to mean and SEM in a contrasting colour, it is hard to see them on the scatterplots.
We have added the number of objects included in each group on each plot in our manuscript, changes have been highlighted in yellow.
Thank you for your answer.
It is recommended to state the numbers of the analyzed cells under the legends of the panels corresponding to the graphs (e.g., Figures 1B-E, 2B-F, 3B, 4B, etc.). Additionally, please include these numbers directly on the graphs for clarity.
Comments 4: Better describe the components used for analyzing different properties: explain what DCFHDA and gH2AX are; describe why red/green ratio is measured for the JC-1 probe, explain how the genes were picked for the qPCR, etc.. Also, provide references to publications demonstrating the relevance of the studied parameters to the oocyte/blastocyst quality.
Response4:
Thank you for your kind suggestion.
(1) Dichlorofluorescein diacetate (DCFHDA) is the oxidationsensitive fluorescent probe to detects intracellular reactive oxygen species levels, we have added the explanation in line 186-187.
(2) γ-H2A.X is the key regulator of DNA damage, we have added the explanation in line 198.
(3) In JC-1 fluorescent test, yields green or red fluorescence, to distinguish cells with high or low mitochondria potential. The mitochondria with a high membrane potential fluoresced red, whereas those with a low membrane potential fluoresced green. We have added the explanation in line 264-266.
(4) SOD1, CAT, and GPX4 are antioxidant-related genes,we tested these genes in order to further proof whether UA supplementation can restored antioxidant capacity of in H2O2-treated porcine oocytes following oxidative stress.
Mitochondrial function related genes AMPKα1, SIRT1 and PGC-1α were tested to explore the effect on mitochondrial function of UA supplementation.
Autophagy helps to remove damaged mitochondria and restore mitochondrial function, so we tested the expression of autophagy related genes (Beclin, LC3B and p62) in order to better explore the potential mechanisms of UA supplementation to enhance mitochondrial function.
The mRNA levels of genes related to the developmental potential of parthenogenetic blastocysts (Nanog and Sox2) and apoptosis (Bax and Bcl2) were added to measure, in order to better proof UA supplementation effect in enhancing developmental quality of parthenogenetic embryos derived from H2O2-treated porcine oocytes.
(5) Relative references to publications demonstrating the relevance of the studied parameters to the oocyte/blastocyst quality have already added in line 57.
Thank you for your answer. I do not have any further comments here.
Comments 5: Fig.5: clarify the parameters for counting aberrant spindles and misaligned chromosomes.
Response5:
Thank you for your kind suggestion. The parameters for counting aberrant spindles and misaligned chromosomes have added in Figure 5A.
Thank you for your answer.
I can see examples of aberrant spindles in Figure 5A, but there is no description of the parameters used to differentiate between normal and aberrant spindles or the criteria for counting chromosomes as misaligned. Please add the spindle assessment parameters and chromosome misalignment criteria to the Materials and Methods section.
Comments 6: In some figures (e.g., Fig. 1D), the axis titles are too general. Make all axis titles descriptive so readers do not need to search the figure legend for explanations.
Response6:
Thank you for your kind suggestion. We have added descriptions to the title of the statistical graph for a better understanding of our result, such as Figure 1D, Figure 2C, Figure 3C, Figure 4B, Figure 6F andG, Figure 8B and Figure 9C and D.
Thank you for your answer.
Please check: Fig. 3C, Fig.7F;G, etc.: should be “Normalized mRNA expression”; Fig.9B: “Total cell number per blastocyst”.
Comments 7: Improve figure quality: In Fig. 7A, LCA-FITC is not visible; in Fig. 8A, Hoechst is barely seen.
Response7:
Thank you for your kind suggestion. We have tried our best to obtain high quality fluorescence images by a laser scanning confocal microscope, the images in Fig. 7A and Fig. 8A may have become blurred due to the compression of the file when uploading the manuscript. We rearranged the PDF of all the images and uploaded them together with the revised manuscript.
Thank you for your answer.
To improve visibility for Figure 7A, the authors could consider using a black-and-white palette if increasing the contrast is not possible. The remaining images in the revised version of the manuscript are of satisfactory quality, and I have no further comments on them.
Comments on the Quality of English LanguagePoint 1: Check spelling and grammar: For example, there is a spelling mistake in the title and in the axis title of Fig. 2E.
Response 1:
Thank you for your kind suggestion. The spelling mistake of “blastocyst” in the title and in the axis title of Fig. 2E has corrected.
Thank you for your answer.
The manuscript title in the present form still contains a grammar mistake.
Author Response
|
3. Point-by-point response to Comments and Suggestions for Authors
|
|
Comments 1: On lines 143-144 you state:
“Our results showed that addition of UA could promote the conventional IVM rates and subsequent embryonic development of porcine oocytes(Table S7)”.
This text does not accurately convey the findings presented in Table S7. Table S7 shows that the addition of 20 µM UA to the IVM medium does not have an effect on PBE and embryo cleavage but increases the blastocyst rate from ≈31% to ≈36%. It should be noted, though, that the beneficial effect of this addition cannot be verified without analyzing blastocyst quality, similar to the approach in Figure 9. Other concentrations of UA either do not have any effects or reduce fitness.
The data presented in Table S7 is of significant interest, as it shows the potential of UA treatment to improve porcine blastocyst maturation and the scope of the expected effect. Therefore, this data should be incorporated into the main text of the article, either as a table in its current form or as a graph, and the results should be discussed in the Discussion section.
The work in the present form demonstrates the protective effect of the UA only after oxidative stress induced by H2O2. Therefore, the title needs to be revised to clearly convey this, for example: “Urolithin A Protects Porcine Oocytes from Artificially Induced Oxidative Stress Damage, Enhancing Oocyte Maturation and Subsequent Embryo Development”.
Note that the Table S7 has a mistake, the heading of the first column should be Concentration of UA , not of H2O2. |
|
Response 1: |
|
Thank you for your comments. The results about the effects of different dose of Urolithin A supplements on porcine conventional oocytes IVM and parthenogenetic embryos developmental competence have added in main text of revised manuscript (line143-147; line380-387). The title has been changed into” Urolithin A Protects Porcine Oocytes from Artificially Induced Oxidative Stress Damage to Enhances Oocyte Maturation and Subsequent Embryo Development”. (line1-2) The first column of Table S7 has changed into “Concentration of UA (μM)”.
Comments 2: Although it's regrettable that the authors cannot offer a more detailed analysis of blastocysts under UA protection except for the apoptotic markers, the challenges are understandable. I have no further comments here. |
|
Response 2: Thank you for your affirmation.
Comments 3: In lines 547-548 you state: “…one-way ANOVA applied where appropriate”; but it is not clear which statistical test was used when it was not appropriate?
Your data has a hierarchical structure, so Nested ANOVA would be more appropriate to use. For example, in the comparison shown in Figure 3B, the DCFHDA fluorescence intensities are the dependent variable that should be analyzed in relation to the main factor (treatment type) and also the nested factor (independent experiments). Please note that in some cases (for example, Figure 7B), you might lose statistical significance; please adjust the text accordingly if that is the case. It would be particularly interesting to see if the significance criteria are met when re-analyzing the data for blastocyst IVM in the presence of 20 uMUA, presented in Figure S7. Please make the lines corresponding to mean and SEM in a contrasting colour, it is hard to see them on the scatterplots. It is recommended to state the numbers of the analyzed cells under the legends of the panels corresponding to the graphs (e.g., Figures 1B-E, 2B-F, 3B, 4B, etc.). Additionally, please include these numbers directly on the graphs for clarity.
Response3: Thank you for your comments. (1) We share similar views with you and consider that Nested ANOVA is more appropriate for analysis in the following studies: First, the effects of different concentrations of H2O2 on the porcine oocyte maturation and parthenogenetic embryo developmental competence; Then, the effects of different dose of Urolithin A supplement on porcine conventional oocytes IVM and parthenogenetic embryos developmental competence.; Finally, the effects of different concentrations of UA supplementation on the porcine oocyte maturation and parthenogenetic embryo developmental competence. We have changed the data analysis methods in sections 2.1 and 2.2 with Nested ANOVA. But considering that the specificity of our experimental materials (oocyte), there had differences in the parameters of fluorescence image shooting between different batches of experiments in order to ensure the visibility of immunofluorescence images. However, we ensured that the parameters of the same batch of oocytes are consistent when conducting fluorescence analysis of each group. We used the same batch of fluorescence images for fluorescence value statistics, and the experiment was repeated for more than 3 times. So one-way analysis of variance (ANOVA) maybe more suitable for the rest data analysis. The specific statistical methods have described in Section 4.8(line 566-568). (2) The lines corresponding to mean and SEM in such Figure 4B, Figure 5B, Figure 7C, Figure 8B and E, Figure 9B, Figure 10B have been corrected in revised manuscript. (3) The numbers of the analyzed cells have been added in the legends and the graphs, such as Figure 1B, D, E and F; Figure 2; Figure 3B-E; Figure 4B; Figure 5B; Figure 6C, D; Figure 7C; Figure 8B and E; Figure 9B; Figure 10B.
Comments 4: I do not have any further comments here. Response4: Thank you for your affirmation.
Comments 5: Fig.5: I can see examples of aberrant spindles in Figure 5A, but there is no description of the parameters used to differentiate between normal and aberrant spindles or the criteria for counting chromosomes as misaligned. Please add the spindle assessment parameters and chromosome misalignment criteria to the Materials and Methods section. Response5: Thank you for your kind suggestion. The description of the parameters used to differentiate between normal and aberrant spindles or the criteria for counting chromosomes as misaligned have added in section 4.5” Fluorescence Staining” (line524-532).
Comments 6: Please check: Fig. 3C, Fig.7F;G, etc.: should be “Normalized mRNA expression”; Fig.9B: “Total cell number per blastocyst”. Response6: Thank you for your kind suggestion. “Normalized mRNA expression” in figures have corrected (line197 and line 293). “Total cell number per blastocyst” have corrected (line 342).
Comments 7: To improve visibility for Figure 7A, the authors could consider using a black-and-white palette if increasing the contrast is not possible. The remaining images in the revised version of the manuscript are of satisfactory quality, and I have no further comments on them. Response7: Thank you for your kind suggestion. We have tried our best to improve visibility the images of CGs and have changed in our revised manuscript (line308).
|
|
4. Response to Comments on the Quality of English Language |
|
Point 1: The manuscript title in the present form still contains a grammar mistake. |
|
Response 1: |
Thank you for your kind suggestion.
We have tried our best to find and correct the grammar mistakes in titles of our manuscript, such as the title of Figure 1 (line 139), Figure 3 (line 182), Figure 4 (line 197), Figure 5 (line 209).
Round 3
Reviewer 2 Report
Comments and Suggestions for Authors
- Thank you for including the data on UA supplementation on the oocyte quality after convential IVM procedure (Fig.2). Please adjust the lines indicating groups that have significant differences, it is in some cases hard to understand which groups are compared.
- But considering that the specificity of our experimental materials (oocyte), there had differences in the parameters of fluorescence image shooting between different batches of experiments in order to ensure the visibility of immunofluorescence images. However, we ensured that the parameters of the same batch of oocytes are consistent when conducting fluorescence analysis of each group. We used the same batch of fluorescence images for fluorescence value statistics, and the experiment was repeated for more than 3 times. So one-way analysis of variance (ANOVA) maybe more suitable for the rest data analysis..."
Please clarify what you mean here. Does it mean that you display data and ANOVA p-values from only one independent experiment?
The IF experiments are commonly analyzed using Nested ANOVA, as it is designed to handle hierarchical data structures, where one factor is nested within another. In your case, the repeated staining experiments (Exp. 1, 2, 3, etc.) are nested within the treatment groups (control, H2O2 treated, and UA+H2O2 treated), so Nested ANOVA can be applied here. As you pointed in the response, immunofluorescence data involves variation between how staining runs in each experiment and differences between treatment groups, so nested ANOVA is specifically useful here, accounting for both sources of variation.
Please consider re-making your statistical analysis.
Comments on the Quality of English LanguageThe title of the manuscript has a grammar mistake.
Author Response
|
3. Point-by-point response to Comments and Suggestions for Authors
|
|
Comments 1: Thank you for including the data on UA supplementation on the oocyte quality after convential IVM procedure (Fig.2). Please adjust the lines indicating groups that have significant differences, it is in some cases hard to understand which groups are compared. |
|
Response 1: |
|
Thank you for your comments. We have adjusted the lines in the Figure 2 (line183).
Comments 2: Please clarify what you mean here. Does it mean that you display data and ANOVA p-values from only one independent experiment?
The IF experiments are commonly analyzed using Nested ANOVA, as it is designed to handle hierarchical data structures, where one factor is nested within another. In your case, the repeated staining experiments (Exp. 1, 2, 3, etc.) are nested within the treatment groups (control, H2O2 treated, and UA+H2O2 treated), so Nested ANOVA can be applied here. As you pointed in the response, immunofluorescence data involves variation between how staining runs in each experiment and differences between treatment groups, so nested ANOVA is specifically useful here, accounting for both sources of variation.
Please consider re-making your statistical analysis. |
|
Response 2: Thank you for your comments. As we mentioned earlier, to ensure the visibility of immunofluorescence images, there had differences in the imaging parameters across different batches of experiments. Therefore, when performing statistical analysis, data using the same imaging parameters were analyzed together. The statistical results shown in the original manuscript represent the results of an independent experiment batch. Each independent experiment used more than 20 oocytes.
We would like to express our gratitude again for your suggestions on the statistical analysis of our experimental data. We have carefully considered your feedback, and we have reanalyzed the results of all immunofluorescence experiments. A total of three experimental batches were used, with each batch testing more than 20 oocytes, and the results are shown in the Figure 4B(line214), Figure 5B(line227), Figure 7C(line265), Figure 8B and E(line311), Figure 9B(line326) and Figure 10B(line360). The results of immunofluorescence statistical analysis have been updated in the manuscript, such as in the analysis of γ-H2A.X immunofluorescence intensity (line 209), in the analysis of actin immunofluorescence intensity (line258), the analysis of mito tracker immunofluorescence intensity (line280), the analysis of CGs immunofluorescence intensity (line325), the analysis of total cell number per blastocyst(line346).
|
|
4. Response to Comments on the Quality of English Language |
|
Point 1: The manuscript title in the present form still contains a grammar mistake. |
|
Response 1: |
Thank you for your kind suggestion.
We have tried our best to find and correct the grammar mistakes in titles of our manuscript, such as the Section 2.1(line112), Section 2.2(line142), Section 2.3(line201), Section 2.4(line216), Section 2.5(line229), Section 2.6(line248), Section 2.7(line267), Section 2.8(line313), Section 2.9 (line311); the title of Figure 4(line214), Figure 5(line227), Figure 6(line246), Figure 7(line265), Figure 8(line311).
